

# A 600-kyr reconstruction of deep Arctic seawater $\delta^{18}O$ from benthic foraminiferal $\delta^{18}O$ and ostracode Mg/Ca paleothermometry

Jesse R. Farmer[1,2,3], Katherine J. Keller[4,5], Robert K. Poirier[5], Gary S. Dwyer[6], Morgan F. Schaller[7], Helen K. Coxall[8], Matthew O'Regan[8], and Thomas M. Cronin[5]

[1]School for the Environment, University of Massachusetts Boston, Boston, MA, USA
[2]Max Planck Institute for Chemistry, 55128 Mainz, Germany
[3]Department of Geosciences, Princeton University, Princeton, NJ, USA
[4]Department of Earth and Planetary Sciences, Harvard University, Cambridge, MA, USA
10 [5]Florence Bascom Geoscience Center, U.S. Geological Survey, Reston, VA, USA
[6]Division of Earth and Ocean Sciences, Nicholas School of Environment, Duke University, Durham, NC, USA
[7]Department of Earth and Environmental Sciences, Rensselaer Polytechnic Institute, Troy NY, USA
[8]Department of Geological Sciences, Stockholm University, Stockholm, Sweden

*Correspondence to*: Jesse R. Farmer (jesse.farmer@umb.edu)

15 **Abstract.** The oxygen isotopic composition of benthic foraminiferal tests ($\delta^{18}O_b$) is one of the preeminent tools for correlating marine sediments and interpreting past terrestrial ice volume and deep-ocean temperatures. Despite the prevalence of $\delta^{18}O_b$ applications to marine sediment cores over the Quaternary, its use is limited in the Arctic Ocean because of low benthic foraminiferal abundances, challenges with constructing independent sediment core age models, and an apparent muted amplitude of Arctic $\delta^{18}O_b$ variability compared to open ocean records. Here we evaluate the controls on 20 Arctic $\delta^{18}O_b$ by using ostracode Mg/Ca paleothermometry to generate a composite record of the $\delta^{18}O$ of seawater ($\delta^{18}O_{sw}$) from fourteen sediment cores in the intermediate to deep Arctic Ocean (700 – 2700 m) covering the last 600 kyr. Results show that Arctic $\delta^{18}O_b$ was generally higher than open ocean $\delta^{18}O_b$ during interglacials but was generally equivalent to global reference records during glacial periods. The reduced glacial-interglacial Arctic $\delta^{18}O_b$ range resulted in part from the opposing effect of temperature, with intermediate-to-deep Arctic warming during glacials counteracting the whole-ocean 25 $\delta^{18}O_{sw}$ increase from expanded terrestrial ice sheets. After removing the temperature effect from $\delta^{18}O_b$, we find that the intermediate-to-deep Arctic experienced large ($\geq$ 1‰) variations in local $\delta^{18}O_{sw}$, with generally higher local $\delta^{18}O_{sw}$ during interglacials and lower $\delta^{18}O_{sw}$ during glacials. Both the magnitude and timing of low local $\delta^{18}O_{sw}$ intervals are inconsistent with the recent proposal of freshwater intervals in the Arctic Ocean during past glaciations. Instead, we suggest that lower local $\delta^{18}O_{sw}$ in the intermediate-to-deep Arctic Ocean during glaciations reflected weaker upper ocean stratification and more 30 efficient transport of low-$\delta^{18}O_{sw}$ Arctic surface waters to depth by mixing and/or brine rejection.



## 1 Introduction

Climate variability on orbital and suborbital timescales is amplified in polar regions as shown by changes in ocean temperature, sea-ice cover, deep-water formation, ecosystems, heat storage, and carbon cycling (Masson-Delmotte et al., 2006; Miller et al., 2010; Serreze and Barry, 2011; Cronin et al., 2017). Yet, the relationship between the Arctic Ocean and global climate remains poorly understood due in part to poor chronostratigraphic constraints of high-latitude marine orbital-scale records (O'Regan et al., 2008; Marzen et al., 2016; Wang et al., 2018), as well as to challenges in applying traditional geochemical paleoceanographic proxies due to incomplete understanding of the biogeochemistry of Arctic Ocean systems. The global implications of a modern changing Arctic require a better understanding of past glacial-interglacial cycles in the Arctic and their link to orbital insolation cycles, atmospheric $CO_2$, and cryospheric processes.

The oxygen isotopic composition ($^{18}O/^{16}O$, expressed as $\delta$-values in per mille (‰)) of benthic foraminifera (hereafter, $\delta^{18}O_b$) is a widely used tool in isotope stratigraphy and correlation of orbital cycles in open ocean sediments (e.g., Lisiecki and Raymo, 2005; Ahn et al., 2017). Pairing $\delta^{18}O_b$ with an independent paleotemperature estimate (for instance, from benthic microfossil Mg/Ca) potentially removes the temperature effect on $\delta^{18}O_b$, isolating the signature of seawater $\delta^{18}O$ change from $\delta^{18}O_b$ (Lear et al., 2000; Billups and Schrag, 2002; Elderfield et al., 2010; 2012).

Despite these advantages, the use of $\delta^{18}O_b$ for stratigraphic and paleoclimate investigations in marginal seas such as the Arctic Ocean has significantly lagged behind its use in the open ocean. This reflects the interpretive challenges posed by marginal seas, where the histories of local temperature and $\delta^{18}O_{sw}$ can greatly diverge from the global signal (e.g., Vergnaud-Grazzini et al., 1977; Cacho et al., 2000; Bauch et al., 2001; Sagawa et al., 2018). These issues are further compounded in the Arctic Ocean by variable microfossil preservation, poorly constrained chronologies, and low and variable sedimentation rates (e.g., Poirier et al., 2012; Alexanderson et al., 2014). As a result, $\delta^{18}O_b$ has not been routinely measured in previous investigations of Arctic Ocean sediment cores.

Despite these challenges, recent work has highlighted the potential for $\delta^{18}O_b$ applications to the Arctic Ocean. Mackensen and Nam (2014) demonstrated that live epifaunal benthic foraminifera calcify close to equilibrium with the oxygen isotopic composition of Arctic bottom waters, suggesting that $\delta^{18}O_b$ faithfully records the $\delta^{18}O_{sw}$ of Arctic bottom waters. Sediment cores from the central and western Arctic possess microfossil abundances sufficient for paleoceanographic reconstruction of glacial-interglacial cycles (Polyak et al. 2004, 2013; Löwemark et al., 2014; Cronin et al., 2014, 2017; Marzen et al., 2016; Wang et al., 2018). Cronin et al. (2019) presented a preliminary $\delta^{18}O_b$ stack from nine independent sediment cores obtained from the central and western Arctic Ocean spanning the last ~600 kyr (Marine Isotope Stages, MIS 1-13).

Here we interpret downcore $\delta^{18}O_b$ from 14 Arctic Ocean sediment cores alongside constraints on past Arctic bottom water temperature (BWT) provided by magnesium/calcium (Mg/Ca) ratios in the calcite shells of benthic ostracodes (Dwyer et al., 1995; Cronin et al., 2012; 2017, Farmer et al., 2012). From this, we calculate a record of seawater $\delta^{18}O$ ($\delta^{18}O_{sw}$) from the





intermediate-to-deep Arctic Ocean. We interpret the $\delta^{18}O_{sw}$ record in terms of the isotopic effects of global ice volume and drivers of Arctic hydrographic change over the last five glacial-interglacial cycles. Our results show that the Arctic Ocean experienced relatively large (~ 1‰) local $\delta^{18}O_{sw}$ variations coherent with glacial-interglacial cycles. This previously

unrecognized local $\delta^{18}O_{sw}$ variability helps to explain the disagreement between Arctic and open ocean $\delta^{18}O_{b}$ stratigraphies and provides a new dataset for testing hypotheses of Arctic cryosphere-ocean change during the late Quaternary.

## 2 Background

### 2.1 Controls on $\delta^{18}O_{b}$

Fossil benthic foraminifera $\delta^{18}O_{b}$ values are controlled mainly by seawater temperature and local $\delta^{18}O_{sw}$, with the latter

integrating contributions from changes in local hydrography and global terrestrial ice volume (Shackleton, 1967; Waelbroeck et al., 2002):

$$\Delta\delta^{18}O_{b} = \Delta\delta^{18}O_{T} + \Delta\delta^{18}O_{L} + \Delta\delta^{18}O_{IV} \tag{1}$$

where T, L, and IV denote temperature, local hydrography, and ice volume, respectively.

The relationship between ambient seawater temperature ('T' in equation 1) and the fractionation of oxygen isotopes in biogenic calcite tests has been assessed in several paleoceanographic studies (e.g., Shackleton, 1974; Marchitto et al., 2014). At thermodynamic equilibrium, the $\delta^{18}O$ of calcite increases by ~0.20 to 0.25‰ for each degree Celsius (°C) of water cooling (Epstein et al., 1953; Shackleton, 1974; Kim and O'Neil, 1997; Matsumoto and Lynch-Stieglitz, 1999; Ravelo and Hillaire-Marcel, 2007). However, many species of foraminifera precipitate in isotopic disequilibrium with ambient seawater

due to kinetic effects related to hydration of $CO_2$ in seawater, metabolic effects (widely referred to as 'vital effects'), and/or preservation effects related to shell dissolution (e.g., Duplessy et al., 2002; Hoogaker et al., 2010; Zeebe, 2014; Poirier et al., 2021). If these factors can be accounted for using a constant correction, as is typically applied for benthic foraminifera (e.g., Graham et al., 1981; Katz et al., 2003), isotopic calcification temperature can be expressed in a paleotemperature equation that links the oxygen isotope composition of foraminiferal calcite to seawater temperature during precipitation (Shackleton,

1967; Matsumoto and Lynch-Stieglitz, 1999; Ravelo and Hillaire-Marcel, 2007).

In addition to temperature, the growth and decay of ice sheets during Pleistocene glaciations drives changes to $\delta^{18}O_{sw}$ that are expected to be globally uniform on the timescale of ocean mixing (~1000 years) ('IV' in equation 1). Growth of ice sheets during glacial inceptions results in increased $\delta^{18}O_{sw}$, and subsequently a greater incorporation of the heavier $^{18}O$ into

foraminiferal tests relative to $^{16}O$.

Finally, $\delta^{18}O_{sw}$ can also reflect input of waters with different $\delta^{18}O$ signatures because of local hydrography ('L' in equation 1). Changes in the $\delta^{18}O$ of seawater can result from local processes such as sea-ice formation and brine rejection,





precipitation-evaporation balances and processes related to advection. In intermediate to deep waters, the isotope signature
of seawater is primarily a function of water mass flow and mixing (Ravelo and Hillaire-Marcel, 2007 and references therein;
Lisiecki and Stern, 2016). The polar Arctic Ocean is uniquely affected by processes that could modify deep Arctic $\delta^{18}O_{sw}$,
including dense, low-$\delta^{18}O$ brines rejected during sea-ice formation (Hillaire-Marcel and de Vernal, 2008; Meland et al.,
2008; Stanford et al., 2011), and the episodic presence of large ice shelves (e.g., Mercer, 1970; Polyak et al., 2001;
Jakobsson et al., 2010; 2016; Geibert et al., 2021), which putatively had extremely low $\delta^{18}O$ compared to ocean waters (e.g.,
Spielhagen et al., 2022).

**2.2 Study Area: Arctic Oceanography**

Four primary water masses characterize the modern Arctic Ocean with characteristic water depth, temperature, salinity, and
$\delta^{18}O_{sw}$ ranges (Figure 1): the Polar Surface Layer (PSL, ~0-50 m, 0 to − 2 °C, ~ 32 to 34 psu, 0 to < - 4‰), Atlantic Water
(AW, ~ 200 to 1000 m, ~0 to 2 °C, ~ 35 psu, 0.21‰), Arctic Intermediate Water (AIW; 1000-2000 m, ~−0.5 to 0 °C; 34.6 to
34.8 psu, 0.26‰), and Arctic Ocean Deep Water (AODW, 2000-4000 m -0.9 to -0.95 °C, 34.9-34.95 psu, 0.28‰) (Aagaard
and Carmack, 1989; Anderson et al., 1994; Jones, 2001; Rudels et al., 2004; Mackensen and Nam, 2014). Warm AW, which
enters the Arctic through the eastern Fram Strait and the Barents Sea, is separated from colder and fresher surface waters by
a deep halocline (~100 – 200 m). The AL flows along the Eurasian Basin continental shelf, splits into two branches at the
Lomonosov Ridge (LR), with the eastern branch flowing poleward along the eastern LR, and the other branch into the
Amerasian Basin. As AL circulates through the Arctic basin proper, it is transformed into what has been called the Arctic
Circumpolar Boundary Current (Rudels et al., 1999). Eventually, the AL water exits the Arctic through the Fram Strait on
the Greenland Slope.

Our study reconstructs the properties AIW and upper AODW (700-2726 m) in the Amerasian Basin (including the Canada
and Makarov Basins, Figure 1), which have similar $\delta^{18}O_{sw}$ to, but are slightly (<0.5 °C) warmer than, AIW and AODW in
the Eurasian Basin. This temperature difference may reflect geothermal warming or sinking of plumes along the slope
entraining warmer AL water (Timmermans and Garrett, 2006; see discussion in Rudels et al., 2015). As this modern
temperature difference is small compared to the typical precision of paleotemperature reconstructions (±1.0°C, Farmer et al.,
2012), we interpret our records as representative of Arctic-wide intermediate-to-deep conditions.





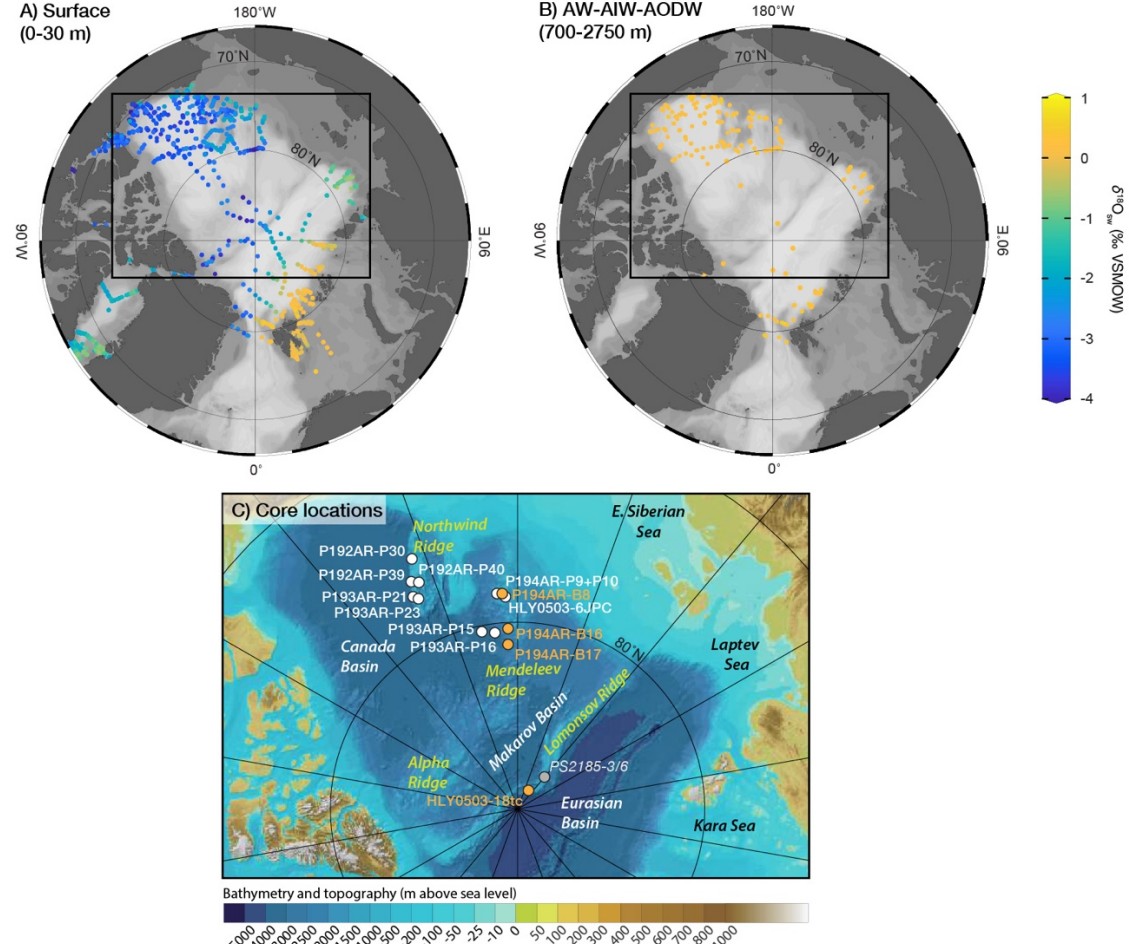

**Figure 1: The oxygen isotopic composition of Arctic Ocean seawater and core locations. Measured $\delta^{18}O_{sw}$ in samples from the surface (0-30 m, panel A) and intermediate-to-deep depths (700-2750 m, panel B) (data compiled in GLODAP2020 version 2, Olsen et al., 2020). Colorbar for (A) and (B) is at right; black box indicates inset of panel (C). (C) Locations of sediment cores used in this**
**study and key geographic features. Basemap for (C) from International Bathymetric Chart of the Arctic Ocean (IBCAO, Jakobsson et al. 2012). (A) and (B) made in Ocean Data View (Schlitzer, 2022). Orange circles denote cores covering the last 50 kyr; white circles denote piston cores contributing to the 600 kyr composite record.**

## 3 Materials and Methods

### 3.1 Materials

This study compiles measured benthic foraminifera oxygen isotopic compositions and ostracode Mg/Ca ratios from ten Arctic piston cores (P1-92-AR-P30, -P39, -P40, P1-93-AR-P21, -P23, P1-94-AR-P9, -P10, -P15, -P16, and HLY0503-06JPC), three boxcores (P1-94-AR-B8, -B16, and -B17), and one trigger core (HLY0503-18tc) (Cronin et al., 2012; 2017; 2019). All cores were recovered from the Northwind and Mendeleev ridges in the western Arctic Ocean at water depths



ranging from 700 to 2726 m, except for HLY0503-18tc, which was recovered from the Lomonosov Ridge in the central

Arctic (Figure 1c).

## 3.2 Geochemical Analyses

### 3.2.1 Oxygen Isotopes

We compiled oxygen isotope measurements from Poore et al. (1999a) and Cronin et al. (2019) made on the benthic foraminifer species *Cassidulina teretis*, *Oridorsalis tener*, and *Cibicidoides wuellerstorfi* (analogous to *Fontbotia*

*wuellerstorfi*, e.g. Wollenburg and Mackensen, 1998; Osterman et al., 1999). Foraminifera were brush-picked, and individual specimens were scored for visual preservation ranging from 1 (transparent) to 4 (visual signs of alteration). Each of these specimens was carefully selected to best avoid any evidence of alteration related to fine-scale dissolution, analogous to the preservation distinctions of Poirier et al. (2021). A minimum of ~30 μg of foraminiferal calcite (~2-8 individual specimens) was used to perform each stable isotope analysis.Analyses for cores AOS94-B8, -B16, and -B17 were performed on a

Finnegan MAT252 with Kiel device at Woods Hole Oceanographic Institution (WHOI) and published by Poore et al. (1999a). Analyses for HLY0503-18tc were performed on a Thermo Delta V+ with Kiel IV device at the Lamont-Doherty Earth Observatory (LDEO). The remaining analyses were conducted at Rensselaer Polytechnic Institute (RPI) using an Isoprime dual-inlet ratio mass spectrometer. LDEO and RPI analyses were first reported in Cronin et al. (2019). All measurements are reported in delta notation relative to Vienna Pee Dee Belemnite corrected to the NBS19 standard, with an

average analytical precision ($1\sigma$) of +/- 0.05 (measurements from LDEO and WHOI) and ($2\sigma$) of +/- 0.08 (measurements from RPI).

### 3.2.2 Ostracode Mg/Ca Paleothermometry

The Mg/Ca of benthic ostracodes *Krithe hunti* and *Krithe minima* (Yasuhara et al., 2014) was previously measured via methods described in Cronin et al. (2012; 2017). Briefly, *Krithe* specimens were brush-picked under a binocular microscope

and assigned a visual preservation index of 1 (transparent) to 7 (opaque white). Only specimens rating 1-5 on this index were measured for Mg/Ca. These specimens were soaked in 5% NaOCl for 24 hours, rinsed five times in high purity deionized water, and dissolved in ultrapure 0.05 N nitric acid. The resulting aqueous solution was analyzed on a Fisons Instrument Spectraspan atomic emissions spectrometer at Duke University.

*Krithe* Mg/Ca was converted to bottom water temperature (BWT) using the linear least squares regional calibration of Farmer et al. (2012):

BWT (°C) = 0.439 * (Mg/Ca$_{K. hunti}$) − 5.14 (2)



Following observations of consistently elevated Mg/Ca in *K. minima* relative to *K. hunti*, *K. minima* Mg/Ca was scaled by 0.77 and converted to BWT using Equation 2 (Cronin et al., 2012; 2017). The prediction error of Equation 2 is ± 1.0°C (Farmer et al., 2012).

The potential for non-thermal factors to bias *Krithe* Mg/Ca have been addressed in detail in previous studies (Cronin et al., 2012; 2017; Farmer et al., 2012) and are summarized here. First, the calibration of *Krithe* Mg/Ca to BWT shows no evidence of a carbonate ion effect when oxidative cleaning is performed (Farmer et al., 2012), in contrast to epifaunal benthic foraminifera (Elderfield et al., 2006). Although other studies note differences in ostracode Mg/Ca when a reductive cleaning step is performed (Elmore et al., 2012; Gray et al., 2014), only oxidative cleaning was performed on the samples used in this study. Second, the high Mg/Ca of glacial- and stadial-aged Arctic *Krithe* cannot be explained by coeval changes in sediment properties. Arctic sediments contain distinct layers of dolomite-rich ice rafted debris (IRD) sourced from Paleozoic sediments in the Canadian Arctic (Bischof and Darby, 1997; Phillips and Grantz, 2001). However, the elevated *Krithe* Mg/Ca intervals occur outside of these discrete dolomitic IRD layers (Cronin et al., 2017). Furthermore, even if dolomite dissolution did occur outside of these intervals, such dissolution could not plausibly raise *Krithe* Mg/Ca. This is because dolomite dissolution releases Mg and Ca in a 1:1 ratio, thereby decreasing seawater Mg/Ca from its current ratio (~5:1). Thus, if significant dolomite dissolution did occur outside of dolomite IRD layers, we would expect this to lower, not raise, *Krithe* Mg/Ca (Cronin et al., 2017).

### 3.3 Chronology

#### 3.3.1 Box and trigger cores (last ~50 kyr)

Chronologies for box cores AOS94-B8, -B16, and -B17 and the HLY0503-18tc trigger core are based on radiocarbon ([14]C) dating on *Neogloboquadrina pachyderma* (Poore et al., 1999a; Hanslik et al., 2010; Poirier et al., 2012). To generate consistent chronologies for these cores, radiocarbon-based age models were recalculated following the approach of Farmer et al. (2021; 2022). Briefly, [14]C dates were recalibrated using Marine20 (Heaton et al., 2020) and assuming no additional marine reservoir age (ΔR=0). Age-depth models were created by Bayesian modeling in rBacon (Blaau and Christen, 2011) using section thickness of 3 cm and accumulation rate prior of 500 years per cm, with all other settings as default. New age-depth models for AOS94-B16 and HLY0503-18tc are provided in the supplementary material.

#### 3.3.2 Piston Cores

The orbital-scale chronology for piston cores is unchanged from the approach detailed in Cronin et al. (2017; 2019). Briefly, an initial age model was created for each piston core using [14]C dates in the upper ~30 cm and two biostratigraphic datums: the benthic foraminifera *Bolivina aculeata,* which has a distinct double spike in abundance dated to Marine Isotope Stage (MIS) 5a, ~ 80 ka (Polyak et al., 2004; Cronin et al., 2014; cf. Hillaire-Marcel et al., 2017), and the planktonic foraminifer



*Turborotalita egelida* abundance peak at ~ 400 ka during MIS 11 (Cronin et al., 2019; O'Regan et al., 2019). From these tie
points, orbital scale age models were established using cyclostratigraphy of benthic calcareous microfossil density
(foraminifera and ostracodes), with intervals of high microfossil abundance aligned to interglacials (Marzen et al., 2016;
Cronin et al., 2019).

One source of uncertainty in these age models is the assignment of the *T. egelida* abundance zone to MIS 11. A recent study
that compared the nannofossil stratigraphy from the Lomonosov ridge (O'Regan et al., 2020) and microfossil
biostratigraphies in a sediment core from the Alpha Ridge, suggested that the *T. egelida* abundance peak could potentially be
older, occurring in MIS 15 or 17 (Vermassen et al., 2021). It should be noted that all cores covering MIS 11 in our age
models source from the Northwind and Mendeleev Ridges in the western Arctic, and not from the Alpha Ridge or
Lomonosov Ridge (Figure 1c). As such, the basin-scale synchroneity of this biostratigraphic marker is uncertain and requires
further work to establish. Moreover, previous multiproxy reconstructions from these western Arctic cores demonstrate
suborbital variability that aligns with independently dated MIS 11 records from the Nordic Seas and North Atlantic (Cronin
et al., 2019). Nonetheless, while consensus exists that microfossil-rich sediments characterize interglacials and interstadials
in the Arctic Ocean (e.g., Poore et al., 1993; Spielhagen et al., 1997; 2004; Polyak et al., 2009; 2013; Hanslik et al., 2013;
Marzen et al., 2016), we acknowledge that the assignment of specific interglacials prior to MIS 5 in Arctic sediments is still
uncertain and may require further stratigraphic refinement. Therefore, we focus our results and discussion primarily on
comparing interglacial and glacial intervals and note that the exact interglacial assignment of features in our records may be
subject to future revision.

### 3.4 Calculation of $d^{18}O_{sw}$

Two approaches were evaluated to calculate $\delta^{18}O_{sw}$. First, *C. wuellerstorfi*-equivalent $\delta^{18}O_b$ was converted to equilibrium
$\delta^{18}O_b$ by subtracting 0.35‰ (Mackensen and Nam, 2014). The $\delta^{18}O_{sw}$ was then calculated from Mg/Ca-based BWT and
equilibrium $\delta^{18}O_b$ following Matsumoto and Lynch-Stieglitz (1999), who derived the following formula from the data of
Kim and O'Neil (1997):

$\delta^{18}O_{sw}$ (‰ VSMOW) = ($\delta^{18}O_{b-eq}$ + 0.27) + 0.2004*BWT + 3.2486        (3)

where $\delta^{18}O_{b-eq}$ is the equilibrium *C. wuellerstorfi* $\delta^{18}O_b$ versus VPDB.


We also calculated $\delta^{18}O_{sw}$ using the linear approximation of Shackleton (1974). In this case, *C. wuellerstorfi*-equivalent
$\delta^{18}O_b$ was converted to *Uvigerina* scale by adding 0.64‰. Then $\delta^{18}O_{sw}$ was calculated using:

$\delta^{18}O_{sw}$ (‰ VSMOW) = ($\delta^{18}O_{b-uvi}$ + 0.27) + 0.25*(BWT – 16.9)        (4)

where $\delta^{18}O_{b-uvi}$ is the *C. wuellerstorfi* $\delta^{18}O_b$ converted to *Uvigernia* scale versus VPBD.






The values of $\delta^{18}O_{sw}$ calculated by Equations 3 and 4 were statistically indistinguishable, differing by 0.05 ± 0.08‰ (2sd, n=353). As such, we present $\delta^{18}O_{sw}$ derived from Equation 3 (using $\delta^{18}O_{b\text{-}eq}$) throughout the text and figures.

Of a total 352 $\delta^{18}O_b$ measurements, 126 $\delta^{18}O_b$ measurements have published Krithe Mg/Ca values and Mg/Ca-derived BWTs
from the same samples (Cronin et al., 2012; 2017). For the remainder of samples, we estimated BWT from an interpolation of the Cronin et al. (2017) intermediate-to-deep Arctic BWT reconstruction.

## 4 Results

The results are separated into two intervals: the last 50 kyr and the entire 600 kyr record. This distinction reflects the different cores used (box and trigger versus piston cores), chronological constraints (radiocarbon versus tuning based on
benthic microfossil density), and data availability (greater in the last 50 kyr) between the two intervals. We first investigate the last 50 kyr to contextualize the longer, albeit more discontinuous records.

### 4.1 Arctic $\delta^{18}O_b$ and $\delta^{18}O_{sw}$: Last 50 kyr

Radiocarbon-dated Arctic $\delta^{18}O_b$ records from AOS94-B8, -B16, -B17 and HLY-0503-18tc are shown with representative extra-Arctic $\delta^{18}O_b$ records in Figure 2. A smoothed spline interpolation to the four Arctic $\delta^{18}O_b$ records fits within ±0.17‰
(root mean square error), suggesting that different Arctic core locations record similar $\delta^{18}O_b$ patterns over the last 50 kyr. However, this Arctic-wide $\delta^{18}O_b$ is noticeably distinct from representative global ocean records particularly since 20 ka. Over the last 10 kyr (approximately the Holocene), Arctic $\delta^{18}O_b$ averages 3.80 ± 0.14‰, versus 2.68 ± 0.10‰ for the global LR04 $\delta^{18}O_b$ stack (Lisiecki and Raymo, 2005), 2.74 ± 0.18‰ in the North Atlantic (CHN82-24, Boyle and Keigwin, 1986), 2.72 ± 0.04‰ in the Southwest Pacific Ocean (ODP 1123, Elderfield et al., 2010) and 2.66 ± 0.25‰ in the Equatorial Pacific
(KNR0734-3PG, Curry et al., 1988) (all $\delta^{18}O_b$ data reported on *C. wuellerstorfi* scale). Conversely, between 35 and 20 ka, a period encompassing late Marine Isotope Stage (MIS) 3 and the Last Glacial Maximum (LGM), Arctic and global records all converge on $\delta^{18}O_b$ values of 4.0 to 4.5‰ (Figure 2a). As a result, open ocean $\delta^{18}O_b$ was 1.5 to 2‰ higher during late MIS 3 and the LGM relative to the Holocene, but Arctic $\delta^{18}O_b$ was only ~0.4 to 0.8‰ higher during late MIS 3 and the LGM relative to the Holocene (Figure 2a).


Based on planktonic foraminifer radiocarbon dates, previous studies have noted that sedimentation in the Arctic Ocean was extremely condensed or possibly absent during the late LGM and early deglaciation (Darby et al., 1997; Poore et al., 1999b; Polyak et al., 2009; Hanslik et al., 2010; Poirier et al., 2012; Jakobsson et al., 2014). In the four Arctic cores used for our composite 50 kyr record, planktonic foraminifer abundance is notably reduced between ~35 and ~14 ka (Figure 2b).
Additionally, only 8 radiocarbon dates fall within this interval, with no dates calibrated to between 21 and 13 ka (Figure 2c). Given this, it is likely that the Arctic $\delta^{18}O_b$ composite does not capture peak glacial conditions ca. 20 ka or the early



deglaciation (Figure 2a). However, this absence would not greatly affect our interpretation, as there is only a modest (< 0.5 ‰) $\delta^{18}O_b$ difference between the peak LGM and late MIS 3 (~35 ka) in both regional records and the LR04 $\delta^{18}O_b$ stack (Figure 2a). To summarize, the potential absence of Arctic benthic foraminifera from peak LGM conditions would not

compromise our ability to capture the majority of the glacial-interglacial $\delta^{18}O_b$ change observed in extra-Arctic records, because Arctic benthic foraminifera from late MIS 3 should capture near-LGM $\delta^{18}O_b$ values.

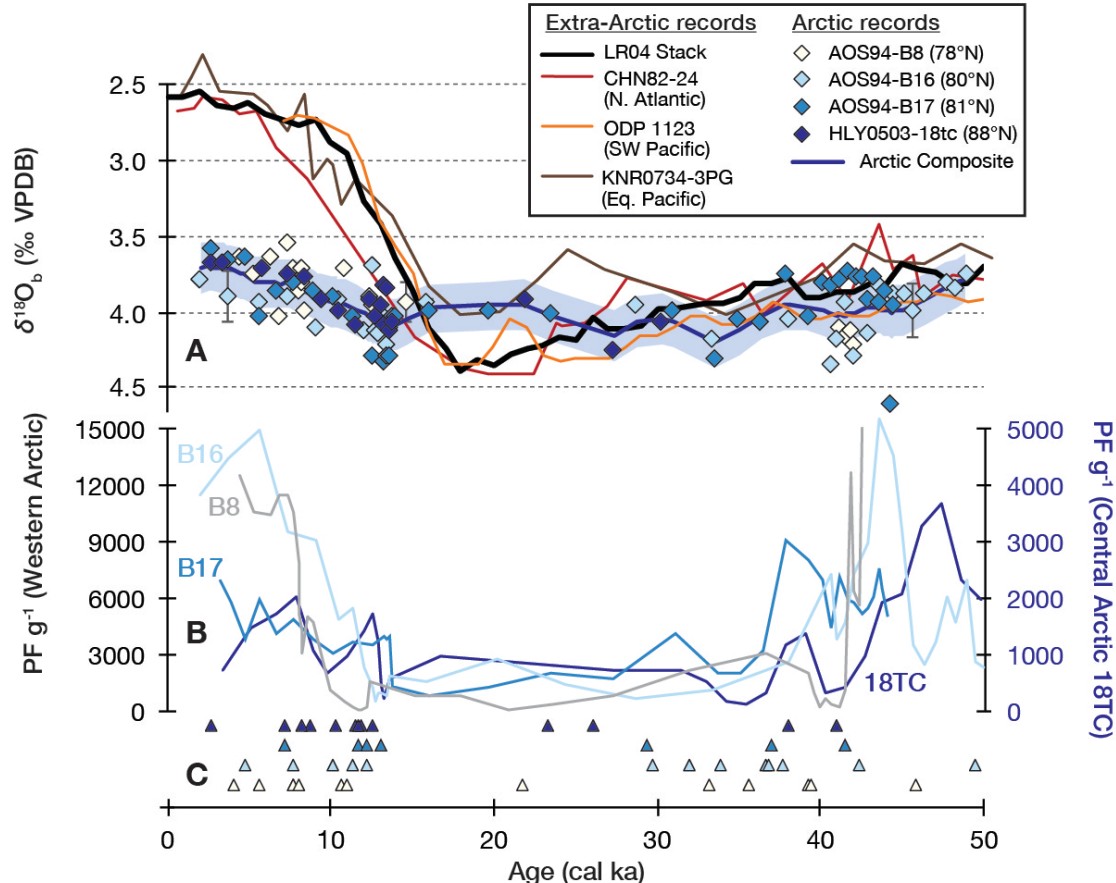

**Figure 2: Chronological constraints on Arctic Ocean sediments over the last 50 kyr and comparison of Arctic $\delta^{18}O_b$ to $\delta^{18}O_b$ from**
**selected deep ocean sites. (A) $\delta^{18}O_b$ data. White, light blue, medium blue, and dark blue diamonds indicate $\delta^{18}O_b$ data points from AOS94-B8, -B16, -B17 and HLY-0503-18tc, respectively. Blue line indicates smoothed spline fit (created in MATLAB using smoothing parameter $p$=0.5); shading indicates ±1 RMSE for spline smoothing function. Other $\delta^{18}O_b$ records shown are the LR04 global stack (black line, Lisiecki and Raymo, 2005), CHN-82-24-4PC (red line, Boyle and Keigwin, 1985); ODP 1123 (orange line, Elderfield et al., 2010), and KNR0734-3PG (brown line, Curry et al., 1988); all records are shown as *C. wuellerstorfi*-equivalent**
**$\delta^{18}O_b$ values. (B) Abundance of planktonic foraminifera per gram of sediment for western Arctic Sites B8 (gray), B16 (light blue) and B17 (medium blue) (left axis), and central Arctic Site HLY0503-18TC (dark blue, right axis). (C) Radiocarbon dates on planktonic foraminifera; colors the same as (A).**

Since both the Arctic and extra-Arctic $\delta^{18}O_b$ should have experienced the same contribution from ice volume changes (e.g., the same $\Delta\delta^{18}O_{IV}$), differences between Arctic and extra-Arctic $\delta^{18}O_b$ records must reflect different temperature ($\Delta\delta^{18}O_T$)





and/or local ($\Delta\delta^{18}O_L$) effects (Equation 1). Figure 3 deconvolves the contributions to $\delta^{18}O_b$ in the Arctic and ODP Site 1123 in the Southwest Pacific (Elderfield et al., 2010). Site 1123 was chosen for comparison because of its high resolution $\delta^{18}O_b$ and infaunal benthic foraminifera (*Uvigerina*) Mg/Ca-derived BWT records covering the last 600 kyr. At Site 1123, *Uvigerina* Mg/Ca is converted to BWT using the preferred equation of Elderfield et al. (2010).

In addition to differences in $\delta^{18}O_b$ (Figure 3a), reconstructed BWT is also notably different between Site 1123 (Elderfield et al., 2010) and the Arctic (Cronin et al., 2012) since 50 ka (Figure 3b). Today, the bottom waters at Site 1123 are warmer (1.3°C) compared to the intermediate and deep Arctic (-0.3°C). This pattern persisted over the Holocene, with ostracode Mg/Ca-derived BWTs of < 1°C compared to benthic foraminifera Mg/Ca-derived BWTs at Site 1123 of ~2°C. However, this pattern reverses during the late MIS 3-to-LGM interval, with warmer Arctic BWTs (1 to 3°C) in comparison to Site 1123 (-1
to 1°C) (Figure 3b).

       Arctic and Site 1123 $\delta^{18}O_{sw}$ records calculated from the respective $\delta^{18}O_b$ and BWT histories are shown in Figure 3c. In the modern ocean, $\delta^{18}O_{sw}$ in the intermediate-to-deep Arctic Ocean (0.26‰, Figure 1b; Mackensen and Nam, 2014) is elevated over that at Site 1123 (-0.10‰, Adkins et al., 2002). Whereas Arctic $\delta^{18}O_{sw}$ was elevated over Site 1123 in the early
Holocene (up to 1‰), both $\delta^{18}O_{sw}$ records converged before 11 ka and maintained similar $\delta^{18}O_{sw}$ values, albeit slightly more positive in the Arctic, back to 50 ka (Figure 3c).

       To investigate the $\delta^{18}O_L$ contribution to $\delta^{18}O_{sw}$ at each location, we subtracted the estimated mean ocean $\delta^{18}O_{sw}$ change from ice sheet change (from Waelbroeck et al., 2002) from Arctic and Site 1123 $\delta^{18}O_{sw}$ records (Figure 3d). The $\delta^{18}O_L$ record of
Site 1123 is roughly constant, with only a slight $\delta^{18}O_L$ elevation before the LGM (0 to 0.2‰) compared to after the LGM (-0.2 to 0‰). In contrast, the Arctic record shows higher $\delta^{18}O_L$ (0 to 0.4‰) during the Holocene and MIS 3, with a $\delta^{18}O_L$ minimum (-0.2 to 0‰) during the LGM and early deglaciation. We note that the Arctic $\delta^{18}O_L$ minimum falls in the interval where foraminifer abundance and radiocarbon dating are particularly limited (Figure 2a), and the presence of this $\delta^{18}O_L$ minimum should be confirmed with additional data. Nonetheless, these results suggest that, in contrast to its elevated $\delta^{18}O_{sw}$
today, the LGM and early deglacial intermediate-to-deep Arctic contained seawater with an oxygen isotopic composition similar to the deep Southwest Pacific Ocean.



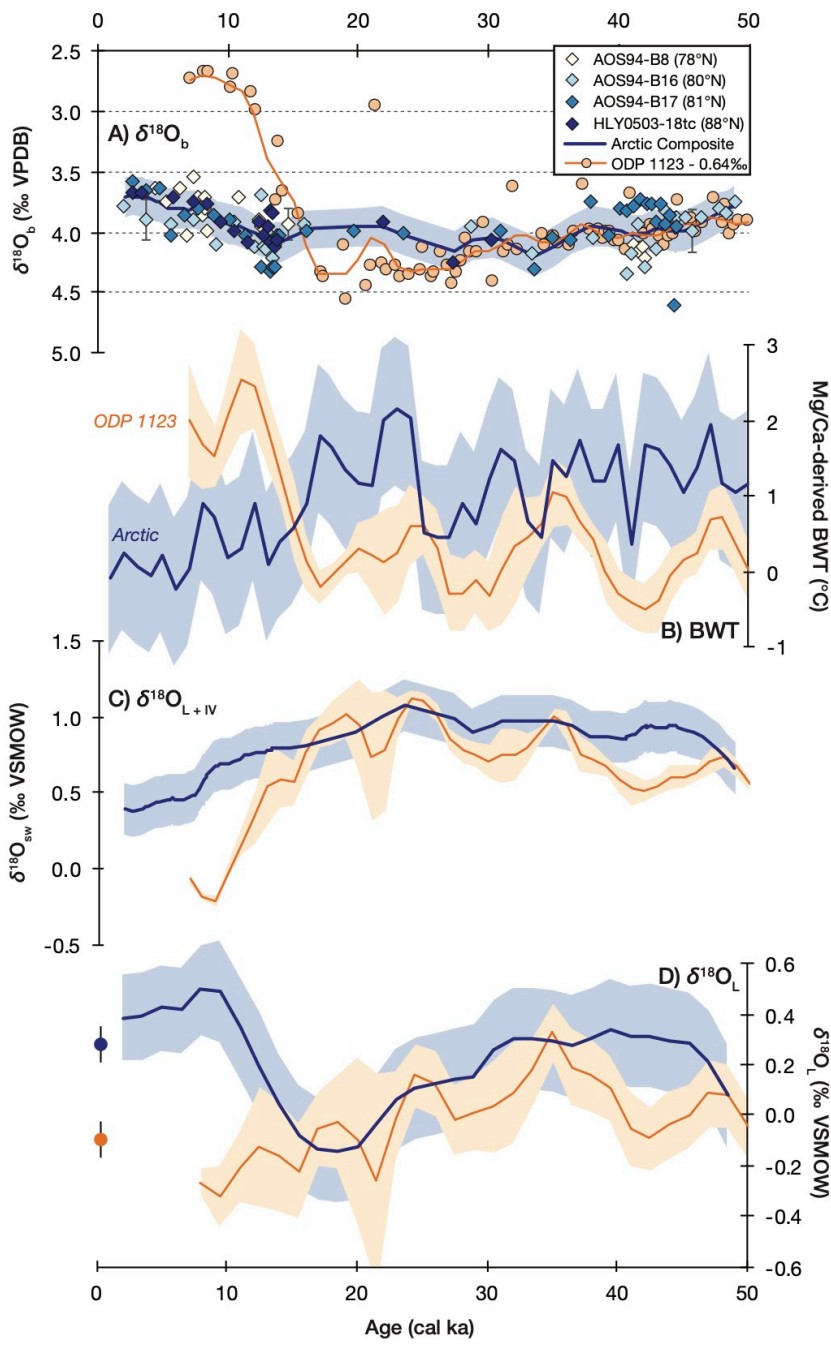

Figure 3: (A) $\delta^{18}O_b$, (B) Mg/Ca-derived BWT, (C) $\delta^{18}O_{sw}$, and (D) $\delta^{18}O_L$ (ice volume corrected) over the last 50 kyr from Arctic
cores (diamonds are individual datapoints; blue lines are composited records) and ODP 1123 (orange circles are individual
datapoints; orange lines are 1 kyr interpolations; Elderfield et al., 2010). Shading denotes propagated 1sd errors. Blue and orange
circle in (D) denote the modern $\delta^{18}O_{sw}$ in Arctic Deep Water and at ODP Site 1123, respectively (Mackensen and Nam, 2014;
Adkins et al., 2002).







## 4.2 Arctic $\delta^{18}O_b$ and $\delta^{18}O_{sw}$: Last 600 kyr

We next assess whether the >0.5‰ of local $\delta^{18}O_{sw}$ variation in the intermediate-to-deep Arctic Ocean over the last 50 kyr also characterized previous glacial cycles. One caveat to calculating Arctic $\delta^{18}O_L$ on these longer timescales is that $\Delta\delta^{18}O_{IV}$ is weakly constrained over older glacial cycles in part because of a paucity of geologic sea-level markers (e.g., Hibbert et al.,
2016). Therefore, instead of calculating Arctic $\delta^{18}O_L$ by subtracting $\delta^{18}O_{IV}$ from Arctic $\delta^{18}O_{sw}$ as we have done for the last 50 kyr, we estimate Arctic $\delta^{18}O_L$ by calculating the $\delta^{18}O_{sw}$ gradient between the Arctic composite and Site 1123 ($\Delta\delta^{18}O_{sw}$) and assume that changes in $\Delta\delta^{18}O_{sw}$ are driven by changing local Arctic $\delta^{18}O_{sw}$.

As for the last 50 kyr, Arctic $\delta^{18}O_b$ typically exhibits a lower range than $\delta^{18}O_b$ at Site 1123 or the LR04 stack over the last
600 kyr. Arctic $\delta^{18}O_b$ was generally higher during interglacials MIS 1, 5, 7, 9, 11, and 13 (Figure 4a) but with generally comparable $\delta^{18}O_b$ in all three records during glacial periods (MIS 4, 8, and 14), with the caveat that peak glacial intervals may not be fully represented in the Arctic $\delta^{18}O_b$ as is the case for MIS 2 (Figure 2a). Additionally, Arctic BWTs demonstrate a similar pattern with respect to Site 1123 BWTs as was observed over the last 50 kyr. Arctic BWTs were similar to or slightly cooler than Site 1123 during interglacials (MIS 5, and putatively MIS 9, 11, and 13 based on our age models; but
curiously not during MIS 7), and generally warmer Arctic BWTs compared to Site 1123 during glacials (MIS 2, 4, 6, and 8). Note the general paucity of benthic foraminifera (and hence $\delta^{18}O_b$ data) in Arctic sediments during peak glacial MIS 6, and putatively MIS 10 and 12 based on our age models, which leads to gaps in our records. The assignment of these gaps to peak glacials results from our age model construction because intervals of low microfossil density are attributed to glaciations (Sect. 3.3 and Marzen et al., 2016).


The $\delta^{18}O_{sw}$ gradient between the Arctic and Site 1123 ($\Delta\delta^{18}O_{sw}$, Figure 4d) averages 0.19 ± 0.35‰ (1sd) over the last 600 kyr, which falls within uncertainty of the modern $\delta^{18}O_{sw}$ gradient of 0.36‰ (Figure 3d). However, this average integrates notable orbital features, with lower $\Delta\delta^{18}O_{sw}$ (-0.7 to 0.2‰) during and/or surrounding glacial stages MIS 2, 6, and putatively 10, 12, and 14, and higher $\Delta\delta^{18}O_{sw}$ (up to 1.0‰) during interglacial stages MIS 1, 5, and putatively 9 and 11. We also
observed pronounced suborbital variability in $\Delta\delta^{18}O_{sw}$, with intervals of $\Delta\delta^{18}O_{sw} < 0$‰ corresponding to MIS 7b and 13b on our age models. There are also multiple $\Delta\delta^{18}O_{sw}$ oscillations of ~1.0‰ within MIS 11, which appear to correspond with MIS 11 substages as previously described (Cronin et al., 2019).




**Figure 4: (A) Individual sample Arctic $\delta^{18}O_b$ (blue diamonds) and 3 point running mean (blue line) compared to interpolated ODP**
**Site 1123 (orange line, Elderfield et al., 2010; 2012) and global ocean LR04 stack (black line; Lisiecki and Raymo, 2005). (B)**
**Reconstructed bottom water temperatures from Arctic ostracode Mg/Ca (blue circles) and 5 point running mean (blue line;**
**Cronin et al., 2012, 2017), and Site 1123 benthic foraminiferal Mg/Ca (orange line). (C) $\delta^{18}O_{sw}$ records from the Arctic (blue**
**diamonds are discrete datapoints; blue line is 3-point running mean) and Site 1123 (orange line; Elderfield et al., 2010; 2012). (D)**
**$\Delta\delta^{18}O_{sw}$ (Arctic – 1123) as a proxy for local Arctic $\delta^{18}O_{sw}$ change (blue diamonds). The modern $\Delta\delta^{18}O_{sw}$ is indicated by the dashed**
**line. Gaps in Arctic record illustrate intervals with limited microfossil abundance. Interglacial marine isotope stages are numbered**
**red; glacial marine isotope stages are shaded gray and numbered; interglacial substages 7b and 13b are shaded blue.**



## 5 Discussion

Our combined ostracode Mg/Ca and benthic foraminifera $\delta^{18}O$ data indicate that the intermediate-to-deep Arctic Ocean experienced large ($\geq 1‰$) changes in local $\delta^{18}O_{sw}$ relative to the Southwest Pacific Ocean over the last five glacial cycles

(Figure 4). The presence of large local $\delta^{18}O_{sw}$ variations is further illustrated in Figure 5. Arctic samples with paired ostracode Mg/Ca-derived BWT and $\delta^{18}O_b$ exhibit a positive correlation between $\delta^{18}O_b$ and BWT (Figure 5a); this is opposite to the trend expected based on inorganic calcite precipitation (Kim and O'Neil, 1997). In contrast, at ODP Site 1123, lower $\delta^{18}O_b$ corresponds with higher Mg/Ca-derived BWT in the same samples (Elderfield et al., 2010; 2012), following the trajectory expected from inorganic calcite precipitation (Figure 5b). Thus, local $\delta^{18}O_{sw}$ variations in the intermediate-to-deep

Arctic Ocean were sufficiently large as to fully overwrite the influence of temperature on benthic foraminiferal $\delta^{18}O$.

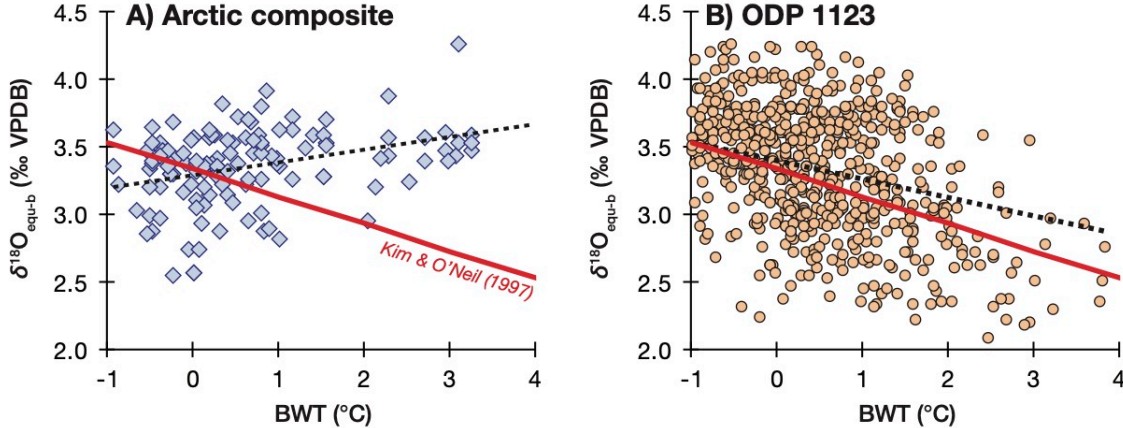

**Figure 5: Paired Mg/Ca-derived bottom water temperature (°C) and equilibrium $\delta^{18}O_b$ (calculated by subtracting 0.35‰ from *C. wuellerstorfi* $\delta^{18}O_b$; Mackensen and Nam, 2014) for the Arctic Ocean (A) and ODP Site 1123 (B). Black dashed line is linear**
**regression to the data; red line is the equilibrium change in calcite $\delta^{18}O$ as a function of temperature (Kim and O'Neil, 1997; Matsumoto and Lynch-Stieglitz, 1999).**

In the last 50 kyr, the lowest local Arctic $\delta^{18}O_{sw}$ was reconstructed between 23 to 14 ka, encompassing the peak of the LGM and the early deglaciation (Figure 3d), but with high uncertainty considering limited sedimentation and foraminifera in this

interval (Figure 2a). Identifying the phasing of previous low local $\delta^{18}O_{sw}$ intervals is even more uncertain given chronological challenges and the absence of $\delta^{18}O_b$ from previous glacial maxima. Nevertheless, we observe a recurring pattern of lower local intermediate-to-deep Arctic $\delta^{18}O_{sw}$ during glacial periods over the last 600 kyr (Figure 4d). This observation requires a source of $^{18}O$-depleted water to the intermediate and deep Arctic during glacial periods (and possibly deglaciations) that was absent during interglacials. We next evaluate several potential candidates for this glacial $^{18}O$-depleted

water source.



### 5.1 Ice shelves and the "Fresh Arctic" hypothesis: A critical evaluation

Based on erosional features along Arctic ridges indicated in geophysical surveys and numerical ice models, it has been argued that large ice shelves existed within the Arctic Ocean during at least some previous glacial maxima (Polyak et al., 2001; Jakobsson et al., 2010; 2016; Niessen et al., 2013; Nilsson et al., 2017; Gasson et al., 2018). Recently, Geibert et al.

(2021) proposed that the Arctic Ocean became filled with freshwater, at least up to 2500 m water depth, during two intervals assigned to MIS 4 (62 – 70 ka) and 6 (131 – 151 ka). The evidence for this "Fresh Arctic" hypothesis comes from the absence of excess $^{230}$Th in multiple Arctic sediment cores ranging from 1000 to 2700 m depth. Because $^{230}$Th is sourced from the radioactive decay of uranium, a conservative element in seawater, Geibert et al. (2021) interpreted the absence of excess $^{230}$Th to represent the absence of uranium in Arctic waters, which could be explained by the displacement of

intermediate-to-deep Arctic seawater with freshwater in these glacial intervals. Additionally, the absence of excess $^{230}$Th corresponds with the absence of cosmogenic $^{10}$Be in several cores, suggesting that the absence of excess $^{230}$Th was coeval with intervals where the Arctic Ocean was shielded from the input of cosmogenic nuclides, potentially by circum-Arctic ice shelves (Geibert et al., 2021).

The "Fresh Arctic" hypothesis (Geibert et al., 2021) has generated significant discussion about both the interpretation of excess $^{230}$Th and cosmogenic $^{10}$Be in Arctic sediments (Hillaire-Marcel, 2022; Geibert et al., 2022a) and the evidence supporting or refuting this hypothesis from outside the Arctic Ocean (Spielhagen et al., 2022; Geibert et al., 2022b). Importantly, any significant contribution of high latitude, $^{18}$O-depleted freshwater should be readily traceable in Arctic $\delta^{18}O_{sw}$, provided that our signal carriers (benthic foraminifera and ostracodes) were present to trace intervals of freshwater

input. Importantly, the foraminifera and ostracodes need not be coeval with the freshwater event. For comparison, during the last deglaciation, a flooding event from the Mackenzie River estimated to last < 1000 years (Keigwin et al., 2018) appeared as a low-$\delta^{18}$O interval covering 5-10 cm of sediment in low sedimentation rate western Arctic cores (Poore et al., 1999b). The amount of freshwater from the deglacial Mackenzie River flooding event would have been orders of magnitude less than would be required to freshen the entire Arctic Ocean. Such a freshening event would conceivably leave a lasting imprint on

$\delta^{18}$O in the low resolution, time-averaged sediments of our study.

We evaluate the coherence of the "Fresh Arctic" hypothesis with our data through two approaches: an isotope mass balance, and an analysis of the temporal trends in our records and the excess $^{230}$Th records of Geibert et al. (2021).

### 5.1.1 Oxygen isotope mass balance

We created a simple mixing model to calculate how Arctic $\delta^{18}O_{sw}$ would change with various contributions of freshwater input from the melting of an ice shelf with $\delta^{18}O_w$ of -20‰ and -40‰ (Figure 6a). The addition of only 1% of freshwater to




the intermediate-to-deep Arctic Ocean would decrease Arctic $\delta^{18}O_{sw}$ by 0.2 and 0.4‰ for a -20‰ and -40‰ freshwater source, respectively, with additional $\delta^{18}O_{sw}$ decline scaling linearly with additional freshwater inputs (Figure 6a).

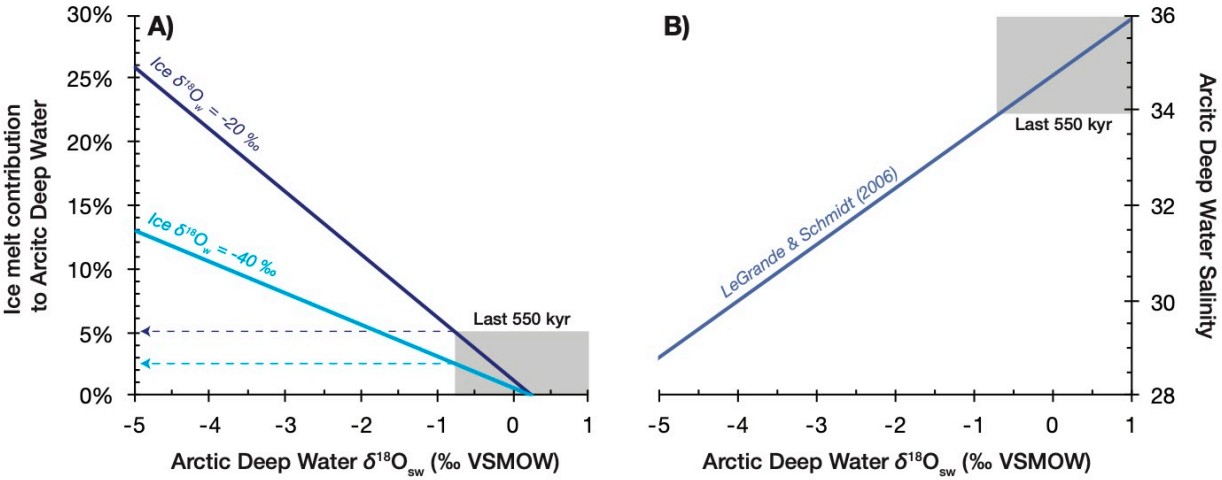


**Figure 6: Two-component mixing models for the $\delta^{18}O_{sw}$ of Arctic Deep Water (A) by mixing with hypothetical ice melt of -20‰ (blue) and -40‰ (cyan), and (B) based on the modern $\delta^{18}O_{sw}$-salinity relationship of the deep Arctic Ocean (LeGrande and Schmidt, 2006). Gray boxes denote the observed range of local Arctic $\delta^{18}O_{sw}$ (from $\Delta\delta^{18}O_{sw}$) in our records (Figure 4d). Dashed arrows in (a) indicate the maximum ice melt contribution needed to explain the lowest Arctic $\delta^{18}O_{sw}$ in our records.**

Our $\Delta\delta^{18}O_{sw}$ reconstruction (Figure 4d) indicates that, over the last 600 kyr, the intermediate-to-deep Arctic Ocean experienced at most a -0.7‰ deviation in local $\delta^{18}O_{sw}$ relative to ODP Site 1123 in the Southwest Pacific when both benthic foraminifera and ostracodes were present in Arctic sediments. Were this local $\delta^{18}O_{sw}$ decline attributed solely to ice melt, this could be explained by at most a 5% contribution of meltwater from a -20‰ ice shelf, or a 2.5% contribution of meltwater from a -40‰ ice shelf, to the modern $\delta^{18}O_{sw}$ of the intermediate-to-deep Arctic Ocean (Figure 6a). Alternatively, considering

the modern $\delta^{18}O_{sw}$-salinity relationship of the deep Arctic Ocean (LeGrande and Schmidt, 2006), a local $\delta^{18}O_{sw}$ decline to -0.7‰ would correspond to a 1.1 unit salinity reduction (Figure 6b). Thus, the local Arctic $\delta^{18}O_{sw}$ variations observed in our 600 kyr records can be explained by only a small percentage contribution of freshwater and are not consistent with the large freshwater volumes proposed by Geibert et al. (2021).

It should be noted that our records exhibit gaps in $\delta^{18}O_b$ data coverage due to absent benthic foraminifera and ostracodes, particularly during intervals ascribed to MIS 6, 10, and 12 on our age models and likely during most glacial maxima. Such occurrence gaps might plausibly be interpreted to represent freshwater intervals (as in Geibert et al., 2021) that excluded marine benthic foraminifera. However, we discount that the existence of these gaps by themselves represent intervals of a fresh Arctic Ocean (at least down to 2500 m water depth) for two reasons. First, the absence of benthic foraminifera and

ostracodes is not uniquely attributable to freshwater. Benthic microfossil abundance is controlled by numerous factors,




including dissolution, ecological preferences such as food availability, and dilution with terrigenous material (e.g., Wollenburg and Kuhnt, 2000; Wollenburg et al., 2004; Scott et al., 2008; Cronin et al., 2013; Polyak et al., 2013; Lazar and Polyak, 2016). In addition, ostracodes inhabit both fresh and marine environments and experience large faunal transitions in many locations, yet no fresh water ostracodes have been observed from Arctic Ocean sediments (Cronin et al. 1995, Poirier

et al. 2012). Instead, terrigenous material dilution provides an explanation consistent with all observations, including low excess $^{230}$Th, low-to-absent $^{10}$Be, and limited microfossil abundance (e.g., Hillaire-Marcel, 2022; but see rebuttal by Geibert et al., 2022a).

Second, even when missing data from glacial maxima, our data proximal to these glacial maxima do not show extremely low

$\delta^{18}O_{sw}$ values, as might be expected from residual freshwater and the time averaging inherent to our low sedimentation rate sites. While our records lack the chronological resolution to speak to millennial-scale events, it is noteworthy that we reconstruct similarly low $\delta^{18}O_{sw}$ values during interglacial substages MIS 7b and 13b as during MIS 6, 10, and 14 (Figure 4d). Provided that this age assignment is correct, the occurrence of low $\delta^{18}O_{sw}$ intervals outside of glaciations rules out an explanation for their cause that is unique to glaciations. As above, these $\delta^{18}O_{sw}$ values can be explained by a small (< 5%)

contribution of freshwater to the deep Arctic and are not consistent with wholesale shifts to a freshwater Arctic Ocean on millennial or longer timescales.

## 5.1.2 Temporal trends in $\delta^{18}O_{sw}$ and excess $^{230}$Th

Geibert et al. (2021) argued that two intervals of low excess $^{230}$Th observed across multiple Arctic Ocean sediment cores date to MIS 4 and 6, and that those intervals arose because freshwater filled the deep basins of the Arctic. We assess the

comparability of our $\delta^{18}O_{sw}$ reconstructions to excess $^{230}$Th data from PS2185-3/6, which shows a similar excess $^{230}$Th pattern observed elsewhere in the Arctic (Geibert et al., 2021). To evaluate this argument, it must be understood that the age assignment from Geibert et al. (2021) for these two intervals is not consistent with the more widely accepted age model published for this core (Spielhagen et al., 2004; see below) or other marine sediment records from this region of the Arctic. The longer-standing and more commonly cited age model would place the two low excess $^{230}$Th intervals observed in PS-

2185 to a thick (70-80 cm) coarse-grained unit deposited during MIS 3 and 4 (Speilhagen et al., 2004; see below). In addition to the excess $^{230}$Th data, previous work on PS2185-3/6 provides planktonic foraminifer abundance (Spielhagen et al., 2004), an indicator of glacial-interglacial changes in Arctic sedimentation (Figure 2; Poore et al., 1993; Polyak et al., 2009; Marzen et al., 2016), as well as neodymium isotopes ($\varepsilon_{Nd}$) of sediment metal oxide coatings, an indicator of the contribution of brines formed on the Eurasian shelves to Arctic intermediate waters (Haley et al., 2008).

timescales.



**Figure 7:** LR04 $\delta^{18}O_b$ stack (A, Lisiecki and Raymo, 2005), local intermediate-to-deep Arctic $\delta^{18}O_{sw}$ (B, this study), and central Arctic site PS2185-3/6 metal oxide $\varepsilon_{Nd}$ (C, Haley et al., 2008), excess $^{230}$Th (D, Geibert et al., 2021), and planktonic foraminifera abundance (E, Spielhagen et al., 2004). The age model for the PS2185-3/6 data (c-e) was based on aligning low excess $^{230}$Th to MIS 4 and 6 as proposed by Geibert et al. (2021). High $\varepsilon_{Nd}$ (plotted downward) is argued to represent increased brine input to the central Arctic (Haley et al., 2008). Gray bars indicate glacials MIS 2, 4, and 6; red bars indicate interglacials MIS 1, 3, and 5a, 5c, and 5e.





Since Geibert et al. (2021) presented their results against core depth, we first created an age model for PS2185-3/6 that aligns
the two intervals of low excess $^{230}$Th with MIS 4 (62 – 70 ka) and MIS 6 (131 – 150 ka) as they proposed. The PS2185-3/6
$\varepsilon_{Nd}$, $^{230}$Th, and planktonic foraminifera abundance data are plotted on the Geibert et al. (2021) age model in Figure 7.
Although this age model aligns the excess $^{230}$Th minima to MIS 4 and 6 (Figure 7d), it creates unrealistic interpretations for
the other datasets from this core. First, the Haley et al. (2008) $\varepsilon_{Nd}$ record shows an extended period of high $\varepsilon_{Nd}$ throughout
MIS 4, 5, and 6 with this age model (Figure 7c). Based on the interpretation of Haley et al. (2008), this would indicate high
brine input to the central Arctic during interglacial MIS 5c, an unusual situation considering the absence of brines during the
Holocene and MIS 3. It also removes any glacial-interglacial variability from the $\varepsilon_{Nd}$ data, leaving the low $\varepsilon_{Nd}$ of the
Holocene through MIS 3 as unique over the past 200 ka. More concerningly, the Geibert et al. (2021) age model requires that
planktonic foraminifera were completely absent from PS2185-3/6 during MIS 5 except for a small abundance (35 planktonic
foraminifera per g sediment) in MIS 5a (Figure 7e). Both the general absence of MIS 5 planktonic foraminifera and the
magnitude of this MIS 5a abundance "peak" sharply contrasts with their prevalence during the Holocene and MIS 3 (> 1000
planktonic foraminifera per g sediment) that is independently constrained in time by $^{14}$C dating (Nørgaard-Pedersen et al.,
1998). Considering the well-established link between interglacial periods and carbonate microfossil-rich sedimentation in the
Arctic Ocean (Figure 2a; e.g., Poore et al., 1993; Spielhagen et al., 1997; 2004; Polyak et al., 2009; 2013; Hanslik et al.,
2013; Marzen et al., 2016; Vermassen et al., 2021), Geibert et al. (2021)'s assignment of low excess $^{230}$Th intervals in
PS2185-3/6 with MIS 4 and 6 is unconvincing and disregards other important stratigraphic indicators that underpin the more
widely accepted age model for this core (Spielhagen et al., 2004).

Figure 8 presents the PS2185-3/6 $\varepsilon_{Nd}$, excess $^{230}$Th, and planktonic foraminifera abundance data on the original age model of
Spielhagen et al. (2004) (which was also used by Haley et al., 2008). Notably, with this age model, maxima in planktonic
foraminifera abundance align with the Holocene, MIS 3, and MIS 5 substages a, c, and e (compare Figure 8e with Figure
7e). Additionally, two intervals of higher $\varepsilon_{Nd}$ correspond to MIS 4 and 6 on this age model (Figure 8c), suggesting a greater
contribution of Eurasian shelf-sourced brines to the intermediate-to-deep Arctic Ocean during glacial maxima as originally
proposed (Haley et al., 2008). With respect to excess $^{230}$Th, the Spielhagen et al. (2004) age model compresses (in time) the
two intervals of low excess $^{230}$Th, with one approximately at the onset of, and the second during, MIS 4 (Figure 8d). It is
notable that our $\delta^{18}O_{sw}$ reconstruction shows values indistinguishable from modern $\delta^{18}O_{sw}$ during this interval (Figure 8b),
which presumably would rule out even a small freshwater contribution to the Arctic Ocean at this time. However, we caution
that the composite Arctic $\delta^{18}O_{sw}$ record does not have sufficiently well-constrained chronology or high enough sedimentation
rates to resolve millennial-scale features like the two low excess $^{230}$Th excursions in MIS 4 (Figure 8d). Still, if the two low
excess $^{230}$Th excursions in MIS 4 are related to large freshwater inputs to the Arctic Ocean, these freshwater intervals must
have dissipated sufficiently rapidly so as not to prevent the accumulation of excess $^{230}$Th within ~1000 years after their
occurrence, and also to leave no apparent trace in Arctic $\delta^{18}O_{sw}$ on multi-millennial timescales.





**Figure 8: As in Figure 7, but with PS2185-3/6 data plotted on the age model of Spielhagen et al. (2004). LR04 $\delta^{18}O_b$ stack (A, Lisiecki and Raymo, 2005), local intermediate-to-deep Arctic $\delta^{18}O_{sw}$ (B, this study), and central Arctic site PS2185-3/6 metal oxide $\varepsilon_{Nd}$ (C, Haley et al., 2008), excess $^{230}Th$ (D, Geibert et al., 2021), and planktonic foraminifera abundance (E, Spielhagen et al., 2004). High $\varepsilon_{Nd}$ (plotted downward) is argued to represent increased brine input to the central Arctic (Haley et al., 2008). Gray bars indicate glacials MIS 2, 4, and 6; red bars indicate interglacials MIS 1, 3, and 5a, 5c, and 5e. Note the alignment of LR04 $\delta^{18}O_b$ stack minima (A) with PS2185-3/6 planktonic foraminifera abundance maxima (E).**

500



## 5.2 Other explanations of reduced glacial Arctic $\delta^{18}O_{sw}$

The lower intermediate-to-deep Arctic $\delta^{18}O_{sw}$ during glacials could reflect a decrease in the supply of high-$\delta^{18}O_{sw}$ Atlantic waters relative to low-$\delta^{18}O_w$ freshwater sources. While both BWT reconstructions (Cronin et al., 2012; 2017) and foraminifera-bound nitrogen isotopes (Farmer et al., 2022) indicate the continuous presence of Atlantic waters in the Arctic Ocean at least since 50 ka, neither technique is sensitive to the rate of Atlantic water inflow. However, data arguing against this hypothesis come from Hoffmann et al. (2013), who demonstrated consistent rates of $^{231}$Pa export from the Arctic Ocean over the last 35 kyr. These observations would putatively require similar rates of Atlantic inflow between the late MIS 3-to-LGM interval and the Holocene, over which local Arctic $\delta^{18}O_{sw}$ varied by ~0.7‰ (Figure 3d). Based on this, we discount that lower glacial intermediate-depth Arctic $\delta^{18}O_{sw}$ reflected reduced input of Atlantic waters during glacial stages.

Instead, our preferred explanation is that the $^{18}$O-depleted water in the intermediate and deep Arctic Ocean reflected stronger mixing between the surface and deep Arctic Ocean. Today, relatively fresh Polar Surface Waters (PSW) possess a low $\delta^{18}O_{sw}$ (of ≤ -1‰) from the input of $^{18}$O-depleted river inflow and Bering Strait inflow (Figure 1a; e.g., Bauch et al., 1995). These low $\delta^{18}O_{sw}$ waters are restricted to the surface by the strong density stratification between the PSW and underlying Atlantic Water that prevents downward mixing of these low $\delta^{18}O_{sw}$ waters to intermediate depths. Today, even the densest, low-$\delta^{18}O_{sw}$ brines that are rejected during sea-ice formation on the shelves are not sufficiently dense to penetrate to intermediate depths (Bauch and Bauch, 2001; Mackensen and Nam, 2014).

During the glacial periods, however, the lower intermediate-to-deep Arctic $\delta^{18}O_{sw}$ may have resulted from weaker upper Arctic density stratification. Such a stratification breakdown has been suggested based on lower foraminifera-bound nitrogen isotope ratios (Farmer et al., 2021), subsurface warming (Cronin et al., 2012), and ostracode faunal assemblages (Poirier et al., 2012) during the late MIS 3-to-LGM interval compared with the Holocene. This weakened glacial stratification could have contributed to lower intermediate-to-deep $\delta^{18}O_{sw}$ by allowing for enhanced vertical mixing directly, or alternatively by facilitating the transport of low-$\delta^{18}O_{sw}$ brines to intermediate depths. We note that we do not observe a consistent relationship between low $\delta^{18}O_{sw}$ intervals and intervals of high sediment leachate $\varepsilon_{Nd}$ at Site PS2185-3/6, which were previously interpreted as evidence for Eurasian shelf brine input to the intermediate-depth Arctic Ocean (Haley et al., 2008). High $\varepsilon_{Nd}$ appears to correspond with low $\delta^{18}O_{sw}$ during MIS 6, but high $\varepsilon_{Nd}$ during MIS 4 aligns with $\delta^{18}O_{sw}$ values that are indistinguishable from the modern (Figure 8), with the caveat that the $\delta^{18}O_{sw}$ record is likely missing peak glacial conditions (e.g., Figures 2 and 4). Nonetheless, given the ubiquity of low-$\delta^{18}O_{sw}$ waters in the upper Arctic Ocean today (Figure 1), any greater contribution of surface waters to depth could have lowered intermediate-to-deep Arctic $\delta^{18}O_{sw}$. These surface waters need not have been sourced exclusively from the Eurasian shelves, and hence may have had a time variable $\varepsilon_{Nd}$ signature.



## 6 Conclusions

We reconstructed the local $\delta^{18}O_{sw}$ of the intermediate-to-deep Arctic Ocean over the last 600 kyr using measurements of oxygen isotopes in benthic foraminifera and Mg/Ca in benthic ostracodes. Arctic $\delta^{18}O_b$ shows a reduced glacial-interglacial amplitude compared to open ocean $\delta^{18}O_b$ records, with higher Arctic $\delta^{18}O_b$ relative to the open ocean during interglacials and

similar Arctic $\delta^{18}O_b$ values to the open ocean during glacials. This reduced Arctic $\delta^{18}O_b$ amplitude is not altogether surprising considering the unique glacial-interglacial temperature history of the intermediate-to-deep Arctic Ocean, with warmer glacials and cooler interglacials (Cronin et al., 2012; 2017). However, removing the temperature contribution to $\delta^{18}O_b$ shows that the intermediate-to-deep Arctic Ocean also experienced relatively large (~ 1‰) changes in local $\delta^{18}O_{sw}$ across glacial-interglacial cycles, with lower local $\delta^{18}O_{sw}$ during glacials and higher $\delta^{18}O_{sw}$ during interglacials.


In assessing potential explanations of lower intermediate-to-deep Arctic $\delta^{18}O_{sw}$ during glacials, we find that the magnitude of glacial $\delta^{18}O_{sw}$ decline does not require dramatic inputs of freshwater to the Arctic Ocean. Moreover, we find that previous arguments for glacial freshwater events in the Arctic Ocean using [230]Th excess are in part based on age models that are inconsistent with accepted patterns of Arctic sedimentology. A reinterpretation of these age models shows that intervals of

low [230]Th excess do not correspond with low intermediate-to-deep Arctic $\delta^{18}O_{sw}$, raising uncertainty about the causal relationship between [18]O-depleted freshwater input and low [230]Th excess intervals. Instead, we find that intervals of lower intermediate-to-deep Arctic $\delta^{18}O_{sw}$ could be explained by reduced Arctic Ocean stratification during glacial periods, with stronger mixing of low-$\delta^{18}O_{sw}$ surface waters and/or brines to intermediate depths.

Ultimately, our results show that $\delta^{18}O_b$ integrates complex histories of temperature and local $\delta^{18}O_{sw}$ changes in the intermediate-to-deep Arctic Ocean. Over the glacial-interglacial cycles of the last 600 kyr, the $\delta^{18}O$ influences of temperature, global ice volume, and local $\delta^{18}O_{sw}$ on $\delta^{18}O_b$ tend to act in opposing directions within the Arctic Ocean. During glaciations, larger continental ice volume worked to raise $\delta^{18}O_b$, but warming and lower local $\delta^{18}O_{sw}$ in the intermediate-to-deep Arctic Ocean worked to lower Arctic $\delta^{18}O_b$. Consequently, the glacial-interglacial $\delta^{18}O_b$ ranges in the Arctic Ocean are

modest. Considering this modest $\delta^{18}O_b$ range and the limited availability of benthic foraminifera in Arctic sediments, $\delta^{18}O_b$ does not appear to hold the same promise as a chronostratigraphic tool in Arctic sediments as in the open ocean. Instead, paired $\delta^{18}O_b$ and ostracode Mg/Ca paleothermometry provides a new approach for investigating past Arctic $\delta^{18}O_{sw}$ variations, which we suggest will serve as a sensitive integrator of oceanographic and cryosphere changes.






## 7 Data availability

The new data in each figure is provided as individual worksheets within a single Excel spreadsheet in the Supplement. These data will be archived in the publicly accessible PANGAEA database with a citable DOI after review.

## 8 Author contributions

Conceptualization: JRF, KJK, TMC; Data curation: JRF, KJK, RKP, TMC; Formal analysis: JRF, KJK; Funding acquisition: JRF, RKP, MFS, TMC; Investigation: JRF, KJK, RKP, GSD, MFS; Methodology: JRF, KJK; Project administration: JRF, TMC; Resources: all authors; Software: JRF; Supervision: JRF, TMC; Validation: all authors; Visualization: JRF; Writing-original draft preparation: all authors; Writing-review and editing: all authors.

## 9 Competing interests

The authors declare that they have no conflicts of interest.

## 10 Acknowledgements

This study was funded by the USGS Climate Research and Development Program; JRF acknowledges additional support from the Lamont-Doherty Earth Observatory Climate Fund and NSF OCE-2054780. We thank L. Gemery and M. Torresan
of the U.S. Geological Survey for assistance sampling cores (P1-92-AR30, P1-93-AR21, P1-94-AR9, P1-92AR39, P1-92-AR40, and P1-94-AR15), L. Polyak for providing samples from HLY0503-06JPC core, and M. Jakobsson for providing samples from HLY0503-18tc core. We are grateful to H. Dowsett and M. Robinson for their reviews of a draft version of this manuscript. Any use of trade, firm, or product names is for descriptive purposes only and does not imply endorsement by the U.S. Government.

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
