# Peer review of "A 600-kyr reconstruction of deep Arctic seawater $\delta^{18}O$ from benthic foraminiferal $\delta^{18}O$ and ostracode Mg/Ca paleothermometry"

_EGUsphere, 2022_

## Author Comment (AC2)

We thank the Reviewers for their constructive and helpful feedback and support for the manuscript. These reviews brought up challenging but fair points that we will extensively address in our revised manuscript. Please see our responses to their feedback in blue, below.
- Jesse Farmer, on behalf of coauthors Katherine Keller, Robert Poirier, Gary Dwyer, Morgan Schaller, Helen Coxall, Matt O'Regan, and Tom Cronin

**Comment on egusphere-2022-1212**
**Robert F. Spielhagen (Referee)**

Referee comment on "A 600-kyr reconstruction of deep Arctic seawater δ18O from benthic foraminiferal δ18O and ostracode Mg/Ca paleothermometry" by Jesse R. Farmer et al., EGUsphere, https://doi.org/10.5194/egusphere-2022-1212-RC1, 2022

Farmer et al. present a collection of benthic oxygen isotope data from Arctic deep-sea sediment cores, derived from benthic foraminifers. The data are partly new, but several data sets have been published in the last decades. The authors use these data to calculate past seawater d18O, taking into account past changes in global ice volume and bottom water temperature. For the latter they use Mg/Ca data, which are a reliable paleothermometer. The major result is that glacial periods of the last 600 kyr often saw significantly lower values of d18O at the sea floor than interglacials. The authors then discuss possible explanations for this observation and rule out that the Arctic Ocean was filled with freshwater during recent glacials, as proposed by Geibert et al. (2021, Nature). Instead they suggest intensive brine formation during glacials as a process which could have led to a downslope transport of dense, low-d18O waters.

The manuscript is written in excellent English; it is well structured and the figures are illustrative. I suggest to add a table with details on core numbers, geographical coordinates and water depths. This table may be added as a supplement. The reader should not be forced to look up all the core details in various papers. There are some vertical differences in temperature even in the deeper Arctic Ocean and since these differences may affect the d18O in carbonates, water depths of individual cores (e.g., those used for Fig. 2) are critical information (even though there is a temperature correction from Mg/Ca data in the d18O data sets).
**Response:** We appreciate this suggestion and will add a new Table 1 that includes core information, age model source, number of d18O and Mg/Ca samples, and time coverage based on our age models.

Overall, I find this manuscript already in a very mature condition. The Abstract is informative, as is the Introduction which gives various aspects of background information. One or two sentences on the research question(s) tackled in this manuscript would be helpful for the reader to better understand what the authors are aiming at. The work performed is nicely described in the last paragraph of the Introduction, but I guess the authors started with a research question before they performed the data acquisition and collection.

The second chapter gives further background information on the various factors controlling oxygen isotopes in seawater and on the modern oceanographic situation in the Arctic Ocean. All necessary details are presented.

The chapter on Materials and Methods supplies details on the measurements performed, on the core chronology, and on the calculation of d18O of paleo-seawater. The subchapters give all necessary details in a concise manner. In particular I like the subchapter on chronology which clearly states some of the current problems with assigning definite ages to interglacial sediments older than 200 ka. Although the U/Th based age model put forward by Hillaire-Marcel et al. (2017) is at odds with the "conventional" age model of, e.g., Jakobsson et al., 2000, Spielhagen et al., 2004; O'Regan et al., 2008, 2020, it should still be mentioned.
**Response:** We will add additional text concerning the $^{231}Pa_{xs}$ and $^{230}Th_{xs}$ approaches of Hillaire-Marcel et al. (2017) and broader context on the conventional age models to this section:

"Regarding the *B. aculeata* zone, alternative age models based on the extinction of excess $^{231}Pa$ and $^{230}Th$ (~140 ka and 300 ka, respectively) in Arctic sediments assign the *B. aculeata* zone to older ages of MIS 8/7 (Hillaire-Marcel et al., 2017). It should be noted that identifying the depth of excess $^{231}Pa$ and $^{230}Th$ is not trivial in Arctic cores

given stratigraphic and redox-driven alterations in excess $^{231}$Pa and $^{230}$Th (Not and Hillaire-Marcel, 2010; Purcell et al., 2022); for instance, the causes of the absence of their excesses during glacials/stadials are not understood and may be from dilution due to excess terrigenous inputs (Hillaire-Marcel et al., 2022) or changes in the composition of Arctic waters (Geibert et al., 2021). Additionally, it is not currently possible to unequivocally accept age models derived by excess $^{231}$Pa and $^{230}$Th, as there is a major disagreement between these age models and those derived from the documented extinction and appearance of calcareous nannofossils (Jakobsson et al., 2001; O'Regan et al., 2020) and results from optically stimulated luminescence dating of quartz grains on both the central and southern Lomonosov Ridge (Jakobsson et al., 2003; West et al., 2021). Finally, the *B. aculeata* zone has only been dated by excess $^{231}$Pa and $^{230}$Th in one location from the Lomonosov Ridge (PS87/030, Hillaire-Marcel et al., 2017), and not in any of our studied western Arctic cores. Therefore, we maintain the assignment of the *B. aculeata* zone to MIS 5a following previous studies (Polyak et al., 2004; Cronin et al., 2014), but encourage the application of excess $^{231}$Pa and $^{230}$Th extinction to these well-studied western Arctic cores to explore potential age revision of the *B. aculeata* zone."

In the Discussion the authors argue that uncertainties with the chronology (including possible hiatuses like in the LGM) make it problematic to discuss individual periods with d18O minima beyond MIS 6. In the following subchapters they almost entirely concentrate an possible explanations for low glacial d18O of seawater and how these explanations can or cannot be reconciled with the Geibert et al. (2021) hypothesis. While I find the arguments sound and the debate highly interesting, I think the authors miss a chance to comment also on paleoenvironments in previous glaciations. I fully acknowledge that the authors want to be cautious with age assignments beyond MIS 6, but still they should discuss to some greater extent than at present the time back to MIS 15 for which they have collected a nice data set shown in Fig. 4. If they do not do this, one may ask why data from MIS 7-15 are shown at all and why d18O differences between individual stages and substages are laid out in detail in subchapter 4.2.

*Response:* This is a valid point. To address this, we have expanded Section 5.2 to further explain the potential models to explain d18Osw variations throughout the record, and additional data that would help to refine this models.

The debate on a possible "fresh glacial Arctic" explanation for the low glacial d18O of seawater makes up most of the Discussion chapter. I find the arguments given highly plausible, but I have to admit that I am somewhat biased against the Geibert et al. (2021) hypothesis, as demonstrated in our comment on that paper (Spielhagen et al., 2022, Nature). Nevertheless, I think that Farmer et al. have done a good job in collecting various other data speaking against a "fresh glacial Arctic" and discussing these in depth so that their own explanation for low glacial d18O data is left as the most plausible hypothesis.

This explanation is described in the last subchapter of the Discussion (5.2). The authors propose a weakened glacial stratification in the glacial oceans as the most likely cause for low d180 of intermediate to deep waters. They explicitly mention "enhanced vertical mixing" and "the transport of low-δ18Osw brines to intermediate depths". While the latter seems sufficiently clear, considering the previous discussion of eNd values in subchapter 5.1, there is no statement on what may have caused "enhanced vertical mixing" and how and where this may have happened in an ice-covered ocean. This needs to be made clear - otherwise it will remain a "black box" for the readers. I also suggest to include a figure (cartoon) showing the proposed scenario for brine formation.

*Response:* Good point that was also brought up by Reviewer 2. We will clarify and expand upon these mechanisms in a revised Section 5.2 alongside discussing their relevance for pre-MIS 6 intervals.

The Conclusion chapter nicely summarizes the major findings discussed in the previous chapter. It ends with some comments on the suitability of benthic d18O data and ostracode Mg/Ca paleothermometry for future paleoclimate research in the Arctic. In my (not necessarily correct) opinion, the latter point should have been tackled with some pros and cons already in chapter 5 and not just as the last sentences of the manuscript.

*Response:* We understand the reviewer's point here, but we think it best (particularly for readers who might be less familiar with Arctic paleoceanography) to leave this text separated from the discussion, so that it will not get lost within detailed arguments about aspects of Arctic Ocean history.

Specific comments by line numbers

93: Actually, the isotopic change during the transition from sea water to sea ice is only very minor (fractionation factor ~1.003; https://doi.org/10.3189/S0022143000042751) and can be neglected when isotopic changes on glacial-interglacial scales are discussed. However, the d18O/salinity relation in ocean waters can be strongly affected. This can lead to density changes and the sinking of low-d18O near-surface waters to greater depths. Considering these details, the statement in line 93 is too much simplified.
*Response:* This is a good point; we will rephrase this to note the distinction between the minimal isotopic fractionation during sea ice formation itself and the densification of low-d18O surface waters via brine addition that could lead to their sinking.

109: What is "AL"?
*Response:* We will change to Atlantic Water (AW) throughout for consistency.

124: color bar
*Response:* Change will be made.

144: analysis. Analyses
*Response:* Change will be made.

185-186: Since the default R is 550 yr in Marine20 (and was 402 yr in Marine13), it might be worth mentioning the R used in this study.
*Response:* We will specify ΔR=0 here.

186: Blaauw (to be corrected also in the list of references!)
*Response:* We will correct this spelling in both locations.

192: Bulimina aculeata
*Response:* Will be corrected.

200: Lomonosov Ridge
*Response:* Will check that spelling is correct.

282: A temperature of -0.3°C may be correct for intermediate waters (AIW), but deeper waters are colder (-0.9°C; see line 106).
*Response:* Will update to "-0.3 to -0.9°C" to encompass the range of BWTs expected for our core sites.

284: Temperatures are numbers and cannot be "warmer", only higher. Check for other usage of "cooler" and "warmer" BWTs throughout the manuscript!
*Response:* Good catch – we will change all direct references to °C to "higher/lower" while keeping general comparisons of bottom waters as "warmer/cooler".

289-291: I suggest to discuss the differences going from 50 ka to present and not vice versa.
*Response:* We will change the presentation of bottom water temperatures and $\delta^{18}O_{sw}$ here to from oldest to youngest.

346 (and 437): I would regard 7b and 13b as interstadials within MIS 7 and 13, comparable to MIS 5d and 5b within MIS 5. Please note that 5b (which was globally just a colder interval within MIS 5) saw one of the largest glaciations over northern Eurasia in the last 200 ky (Svendsen et al., 2004, QSR).
*Response:* This point will be acknowledged, with the caveat that the Δd18Osw excursions we observe for MIS 7b and 13b greatly exceed those observed in either MIS 5b or 5d.

375: Since freshwater is buoyant, wouldn't it make more sense to say that the Arctic Ocean may have been filled with freshwater DOWN to 2500 m water depth? It certainly depends on the point of view, but I would understand "up to 2500 m" as the description of a bottom water mass.
*Response:* Good point and agreed; we will change to "down to 2500 m water depth".

451: Spielhagen
*Response:* We will correct this spelling (apologies to the reviewer!)

455: timescales???
*Response:* We will delete this typo.

470: Moreover, there were no large ice sheets on northern Eurasia during MIS 5c (Svendsen et al., 2004, QSR) to produce large amounts of brines.
*Response:* This is a good point, but we seek to be cautious here considering the simplicity of the age model. The one Nd isotope datapoint within MIS 5 on this age model dates to 94 ka (within MIS 5c), but the uncertainty on this age estimate is probably significant. We do not think we could rule out that this point overlaps with the expanded Kara-Barents Ice Sheet of the Early Weichselian.

519-521: This is correct. However, there is a recent paper (Rogge et al., 2022, https://doi.org/10.1038/s41561-022-01069-z) showing the formation of plumes which are sinking down to 1200 m even under modern conditions. During glaciations with (partly) ice-covered shelves and strong erosion, conditions allowing the formation of sediment-laden plumes may have occurred even more frequently than during interglacials. Since the Arctic deep-sea basins (>2500 m) are filled with fine-grained sediments (plumites, turbidites; see Goldstein, 1983, DOI: 10.1007/978-1-4613-3793-5_9; Svindland and Vorren, 2002, https://doi.org/10.1016/S0025-3227(02)00197-4), such plume formation may have been a major process for the vertical (and then horizontal) transport of both fines and low-d18O waters during glacials. Maybe the authors want to consider this possibility...
*Response:* This is an interesting observation and we have included the reference to Rogge et al. (2022) in the discussion of potential densification pathways for surface waters. However, we do not wish to place too much emphasis on this mechanism, as it appears (at least to us) difficult to reconcile an increase in sediment-laden plume transport to the deep Arctic during glacial intervals, when glacial sedimentation rates were lower than interglacial sedimentation rates. If anything, we might predict the opposite pattern based on sedimentation rates.

Figures: When figures consist of several "subfigures" (e.g., data panels), they are labeled A, B, C... In the text, they are referenced as a, b, c...
*Response:* We will update panel labels to lower case on figures and in the captions to fit Climate of the Past formatting.

**Comment on egusphere-2022-1212**
**Kaustubh Thirumalai (Referee)**
Referee comment on "A 600-kyr reconstruction of deep Arctic seawater δ18O from benthic foraminiferal δ18O and ostracode Mg/Ca paleothermometry" by Jesse R. Farmer et al., EGUsphere, https://doi.org/10.5194/egusphere 2022-1212-RC2, 2022

Kaustubh Thirumalai, University of Arizona

I found this manuscript by Farmer and colleagues to be highly interesting. The authors attempt to reconstruct bottom water d18Osw values in the Arctic using a combination of benthic foraminiferal d18O measurements paired with ostracode Mg/Ca paleothermometry. Using Site 1123 (SW Pacific Ocean) bottom water d18Osw as a reference record, they find that local Arctic d18Osw (corrected for the influence of ice volume-related changes in global oceanic d18O) is lower than the modern difference between these records during glacial periods over the past 600 kyr. A major portion of the discussion in this manuscript focuses on refuting the "fresh glacial Arctic" hypothesis (Geibert et al. 2021) to explain the anomalously lower d18Osw values in the glacial - which is compelling. The authors instead prefer a mechanism that involves "stratification breakdown" wherein relatively lower-d18O upper ocean waters sink to the bottom due to relatively higher densities (via salinity) modulated by brine formation.

Overall this work is compelling, of interest to the broader community, and I feel that the manuscript will be eventually suitable for publication in Climate of the Past pending some revision. My major concerns regarding this version are two-fold: one centers around the discussion and proposed mechanism of brine-formation to explain relatively lower-d18O in Arctic bottom-waters, and the other is a request to assist readers by providing more information on some of the details related to Mg/Ca paleothermometry and background information. I detail my major and minor suggestions to improve this work below:

Mechanisms and "low-d18O" of brine versus "lower d18O" of surface waters:
• The authors refer to "low-d18O" brines (Line 93), but the briny waters themselves are not anomalously lower in their d18Osw values due to relatively low ice-water d18O fractionation (see e.g. Yamamoto et al. 2001, 2002). Thus, I suggest expanding the text in the introduction to discuss how brine potentially affects bottom water d18O by advecting relatively lower upper-ocean d18Osw (Line 93 and thereafter).
*Response:* This is a good point and also picked up by Reviewer 1. We will rephrase L94-97 to note the distinction between the minimal isotopic fractionation during sea ice formation and the densification of low-d18O surface waters via brine addition (and/or sediment entrainment, e.g. Rogge et al., 2022) that could lead to their sinking.

• I ask the authors to explore/discuss the work of Rasmussen and Thomsen (2009) who suggest that the initial thermohaline origin of brine formation can modulate their density and stable oxygen isotope composition. I recognize that the viewpoint from these authors on brine formation as a paleoceanographic driver has been updated since that paper (e.g. Rasmussen and Thomsen, 2014). However, their benthic d18O values in a region of active brine formation in the Barents Sea appear similar (Figs. 2–3 in 2009 paper) to the relatively lower glacial values observed in this manuscript (e.g. Fig. 3A). Perhaps this can be used as support for brine-formation as a potential cause of the underlying data? The authors also discount the possibility of reduced Atlantic water input during glacial stages.
*Response:* We will expand the discussion of brine formation in Section 5.2 and will add a new figure speaking to its potential contribution to lower Arctic d18Osw values, but we note that our discussion of brines is necessarily simple because of the lack of available data constraints. We emphasize that, with the data constraints available, we cannot exclude that brine formation may have contributed to the intermediate-to-deep Arctic d18Osw patterns we observe, nor can we prove that brine formation had any impact on these d18Osw patterns. Our hope is that these d18Osw data provide motivation for the collection of additional datasets (both radiogenic isotope tracers of weathering product input from the shelves and planktic foram-based d18Osw reconstructions, which we will mention in Section 5) that can better refine the role of brine formation in the modification of Arctic deep waters.

• Although the records do not overlap entirely with the one presented in this manuscript, Ford and Raymo (2019) show that d18Osw (not corrected for ice volume) from ~400–600 ka at Sites 607 and 1208 in the North Atlantic and Pacific respectively are relatively similar to that at Site 1123 (Fig. 3 in their paper). Thus, perhaps the authors can use these records across the interval of overlap to more robustly assert that changes in inflow did not cause the anomalously low glacial d18Osw in the Arctic Ocean (e.g. via a mixing model or relative differences between sites)? I recognize that their data points are fewer in this interval and that overlap is not complete, yet, I feel that this exercise would solidify their argument.

*Response:* Thank you for this idea. We will add the Ford and Raymo DSDP 607 d18Osw reconstruction to Figure 4 and assess its comparability in the discussion.

• What is the driver of the stratification breakdown in the Arctic Ocean under glacial conditions? Presumably there was more perennial sea-ice coverage during glacial times, which could perhaps result in more year-round brine formation, but what would cause more mixing given that this would also likely reduce the impact of winds? Given that perennial Arctic sea-ice under pre-industrial conditions also covered a large extent of the basin, what changes during glacial times that cause radically different oceanic structure?

*Response:* This was presented by Farmer et al. (2021), but we erred in not presenting the concept again here and will correct this in Section 5.2. In brief, the loss of freshwater input to the Arctic Ocean under colder glacial climates (from both reduced precipitation as well as river input) leads to a more saline surface mixed layer, breaking down the strong stratification that characterizes the upper Arctic today

• On this note, I wonder whether the "hyperpycnal flows" hypothesis of Stanford et al. (2011) might have a role to play here? Could intensified hyperpycnal flows related to runoff/melt in spring/summer coupled with more brine formation in the fall/winters be a viable way to reduce year-round d18Osw in the water column? Stanford et al. (2011) also reject brine formation as a likely mechanism in the North Atlantic as there are different predictions for planktic versus benthic d18O values. Considering this, I suggest the authors to expand their discussion section to include the implications of (not-yet-generated) planktic foraminiferal derived/surface-ocean d18Osw in the Arctic and how it may help falsify/strengthen their hypothesis.

*Response:* Good point and we will expand this discussion, including assessing how planktic-benthic d18Osw gradients may help clarify the brine situation. Regarding sediment-laden hyperpycnal flows, however, one would expect that if they increased in frequency during the glacial periods to lower intermediate-to-deep Arctic d18Osw, the sedimentation rates in the Arctic cores would have increased. But there is strong evidence that glacial sedimentation rates were much lower (and in some cases sedimentation may have been absent) during glacial intervals.

Discussion surrounding Mg/Ca and its uncertainty
• The authors ought to provide more background information on ostracode Mg/Ca analysis and their underlying uncertainties. Although it is mentioned that a 'Fisons Instrument Spectraspan atomic emissions spectrometer' instrument was used (Lines 157–158), no details are provided about instrument precision, matrix/standard effects, external standard replicability, numbers of valves analyzed, what instaars were utilized, inter-sample variance etc. Although the authors cite previous studies that go more into detail on some of these aspects, I think that these details need to be in this manuscript, where the Mg/Ca data are front and central to the assertions.

*Response:* We will update Section 3.2.2 to provide additional methodological details on the Mg/Ca determination: "Only adult specimens rating 1-5 on this index were measured for Mg/Ca. These specimens were soaked in an oxidative solution of 5% NaOCl for 24 hours to remove any reactable organic material, rinsed five times in high purity deionized water under mild sonication to remove any residual inorganic surface material, dissolved in ultrapure 0.05 N nitric acid and analyzed by atomic emission spectrometry (AES) on a Fisons Instruments Spectraspan 7 AES at Duke University using matrix-matched calibration standards from ultrapure, plasma-grade standards (SPEX brand). Unlike foraminiferal shells, which are also commonly used for marine Mg/Ca-BWT reconstructions, ostracodes produce a smooth, solid, chamber-free shell, making ostracodes far less susceptible to clay particle contamination than foraminiferal shells (Dwyer et al., 2002). Nevertheless, as a precaution, contamination-prone metals (Al, Fe, and Mn) were simultaneously monitored to ensure the absence of any signal for these constituents. External precision on Mg/Ca ratios using this method yields a relative error (1sd) of ≤1.5%

based on replicate analyses of an in-house limestone consistency standard prepared and analyzed along with each batch of ostracode specimens (Dwyer et al., 2002)."

• Accordingly, it is not clear how the "1sd" propagated error in Fig. 3A was constructed - was this propagated through a Monte Carlo procedure? It doesn't seem like a constant "calibration" error of ±1°C (Line 164–165; Farmer et al. 2012) was employed. Thus, a clearer discussion of the procedures employed to propagate Mg/Ca uncertainties into the resultant BWT and d18Osw records is needed.

*Response:* We agree that this was unclear, and for clarity in calculation, error propagation, and comparison to other studies (Sites 607 and 1123), we will recalculate d18Osw for the Arctic record using PSU Solver (Thirumalai et al., 2016). This requires modification to PSU Solver to account for the measurement of Mg/Ca and d18O in different phases, as will be described in Section 3.4. Uncertainty envelopes on BWT and d18Osw shown in the Figures will now all be 1sd-equivalent (32 and 68% confidence intervals) on the Monte Carlo propagated error (n=1000 realizations as default in PSU Solver). Importantly, while employing PSU Solver clarifies our d18Osw error propagation, the absolute d18Osw values have not appreciably changed from the previous manuscript.

• Relatedly, there is no discussion about other trace metal values such as Mn/Ca, Fe/Ca, or Al/Ca in these measurements, which were presumably also measured alongside Mg/Ca for investigating clay contamination. To what extent are these parameters correlated or uncorrelated with the Mg/Ca data and what are the implications for sediment-based or post-depositional alteration?

*Response:* We will specify that Al, Fe, and Mn were monitored as indicators of clay contamination during Mg/Ca analysis. No detectable signal was found for these elements.

Statistical robustness of d18Osw reconstruction:
• Although I am somewhat convinced looking at Fig. 4D that glacials generally have lower d18Osw in the Arctic relative to Site 1123, I feel that this can be done in a more robust manner by compositing glacials and interglacials across the entire record - what is the average value (my apologies if I missed this in the text) of the difference in glacials versus interglacials? What is the impact of highly variable Δd18Osw values in MIS 7, 11 and 13 interglacials on this average, as many of these anomalies appear to be equivalent or lower than the MIS 2–4 glacial values? Moreover, how robust is this average glacial-interglacial difference relative to the modern difference given the propagated uncertainty (see above point on Mg/Ca)? Although the 0.3–0.6 ‰ anomalies are likely significant, I wonder about smaller magnitude anomalies…

*Response:* This was a good point and will be included in the text in Section 4.2. The results of these calculations show that interglacial Δd18Osw is significantly higher than glacial Δd18Osw when composited across all data in this study, using either Site 1123 (P<0.001) or Site 607 (P=0.009) as the normalization for calculating Δd18Osw.

Minor Points:
• It would be helpful to label in the caption that Fig. 4C refers to d18OL+IVC at both of the sites, right? If so, given that the record in Fig. 4D is constructed to be non-dependent on ice-volume/sea-level, I ask whether Fig. 4C is required at all? Moreover, this sub-plot is not cited/discussed explicitly in the text.

*Response:* We made two changes for clarification here: First, we will explicitly define and refer to ice-volume corrected d18Osw (as in Figure 3d) as d18Osw-ivc. Second, we will expand our discussion of Figure 4c in the text to compare Arctic d18Osw with d18Osw from Sites 607 and 1123.

• The authors provide information about the d18O measurements - however, am I to understand correctly that this work only involved collating previously measured d18O data? In that case, I would recommend adding "In these studies" before "Foraminifera were brush-picked," on Line 140 and updating Line 148 to say "In these previous studies, all measurements were reported…" to clarify what has been done in this work versus previous studies.

*Response:* Good point; we will adjust accordingly.

• Line 225: Although the average error and standard deviation was relatively low, I'd recommend providing an estimate of the overall ranges as well (Eqn. 3 estimates a of range from X to Y whereas Eqn. 4)

*Response:* As we have now updated the methods to use PSU Solver, we will remove this comparison from the text.

---

## Author Response (AR1)

We thank the Reviewers for their constructive and helpful feedback and support for the manuscript. We have addressed their comments; references to the updated text and its line numbers within the resubmitted text are provided here.

- Jesse Farmer, on behalf of coauthors Katherine Keller, Robert Poirier, Gary Dwyer, Morgan Schaller, Helen Coxall, Matt O'Regan, and Tom Cronin

**Comment on egusphere-2022-1212**
**Robert F. Spielhagen (Referee)**

Referee comment on "A 600-kyr reconstruction of deep Arctic seawater δ18O from benthic foraminiferal δ18O and ostracode Mg/Ca paleothermometry" by Jesse R. Farmer et al., EGUsphere, https://doi.org/10.5194/egusphere-2022-1212-RC1, 2022

Farmer et al. present a collection of benthic oxygen isotope data from Arctic deep-sea sediment cores, derived from benthic foraminifers. The data are partly new, but several data sets have been published in the last decades. The authors use these data to calculate past seawater d18O, taking into account past changes in global ice volume and bottom water temperature. For the latter they use Mg/Ca data, which are a reliable paleothermometer. The major result is that glacial periods of the last 600 kyr often saw significantly lower values of d18O at the sea floor than interglacials. The authors then discuss possible explanations for this observation and rule out that the Arctic Ocean was filled with freshwater during recent glacials, as proposed by Geibert et al. (2021, Nature). Instead they suggest intensive brine formation during glacials as a process which could have led to a downslope transport of dense, low-d18O waters.

The manuscript is written in excellent English; it is well structured and the figures are illustrative. I suggest to add a table with details on core numbers, geographical coordinates and water depths. This table may be added as a supplement. The reader should not be forced to look up all the core details in various papers. There are some vertical differences in temperature even in the deeper Arctic Ocean and since these differences may affect the d18O in carbonates, water depths of individual cores (e.g., those used for Fig. 2) are critical information (even though there is a temperature correction from Mg/Ca data in the d18O data sets).
*Response:* We appreciate this suggestion and have added a new Table 1 that includes core information, age model source, number of d18O and Mg/Ca samples, and time coverage based on our age models (L131-135).

Overall, I find this manuscript already in a very mature condition. The Abstract is informative, as is the Introduction which gives various aspects of background information. One or two sentences on the research question(s) tackled in this manuscript would be helpful for the reader to better understand what the authors are aiming at. The work performed is nicely described in the last paragraph of the Introduction, but I guess the authors started with a research question before they performed the data acquisition and collection.

The second chapter gives further background information on the various factors controlling oxygen isotopes in seawater and on the modern oceanographic situation in the Arctic Ocean. All necessary details are presented.

The chapter on Materials and Methods supplies details on the measurements performed, on the core chronology, and on the calculation of d18O of paleo-seawater. The subchapters give all necessary details in a concise manner. In particular I like the subchapter on chronology which clearly states some of the current problems with assigning definite ages to interglacial sediments older than 200 ka. Although the U/Th based age model put forward by Hillaire-Marcel et al. (2017) is at odds with the "conventional" age model of, e.g., Jakobsson et al., 2000, Spielhagen et al., 2004; O'Regan et al., 2008, 2020, it should still be mentioned.
*Response:* We have revised the text in Section 3.3.2 to better incorporate alternative age models from excess [231]Pa and [230]Th, while also describing the reasoning for our chosen age model framework (L200-240).

In the Discussion the authors argue that uncertainties with the chronology (including possible hiatuses like in the LGM) make it problematic to discuss individual periods with d18O minima beyond MIS 6. In the following

subchapters they almost entirely concentrate an possible explanations for low glacial d18O of seawater and how these explanations can or cannot be reconciled with the Geibert et al. (2021) hypothesis. While I find the arguments sound and the debate highly interesting, I think the authors miss a chance to comment also on paleoenvironments in previous glaciations. I fully acknowledge that the authors want to be cautious with age assignments beyond MIS 6, but still they should discuss to some greater extent than at present the time back to MIS 15 for which they have collected a nice data set shown in Fig. 4. If they do not do this, one may ask why data from MIS 7-15 are shown at all and why d18O differences between individual stages and substages are laid out in detail in subchapter 4.2.

*Response:* We have revised Section 5.2 to be inclusive of previous glacial cycles (L497-546) and also added a new Section 5.3 that raises some of the older intervals as potential targets for multiproxy reconstructions to assess the cause(s) of lower d18Osw during glacials (L548-576).

The debate on a possible "fresh glacial Arctic" explanation for the low glacial d18O of seawater makes up most of the Discussion chapter. I find the arguments given highly plausible, but I have to admit that I am somewhat biased against the Geibert et al. (2021) hypothesis, as demonstrated in our comment on that paper (Spielhagen et al., 2022, Nature). Nevertheless, I think that Farmer et al. have done a good job in collecting various other data speaking against a "fresh glacial Arctic" and discussing these in depth so that their own explanation for low glacial d18O data is left as the most plausible hypothesis.

This explanation is described in the last subchapter of the Discussion (5.2). The authors propose a weakened glacial stratification in the glacial oceans as the most likely cause for low d180 of intermediate to deep waters. They explicitly mention "enhanced vertical mixing" and "the transport of low-$\delta18Osw$ brines to intermediate depths". While the latter seems sufficiently clear, considering the previous discussion of eNd values in subchapter 5.1, there is no statement on what may have caused "enhanced vertical mixing" and how and where this may have happened in an ice-covered ocean. This needs to be made clear - otherwise it will remain a "black box" for the readers. I also suggest to include a figure (cartoon) showing the proposed scenario for brine formation.

*Response:* Good point that was also brought up by Reviewer 2. We have clarified the discussion of the vertical mixing vs. brine formation mechanisms in Section 5.2 (L497-546).

The Conclusion chapter nicely summarizes the major findings discussed in the previous chapter. It ends with some comments on the suitability of benthic d18O data and ostracode Mg/Ca paleothermometry for future paleoclimate research in the Arctic. In my (not necessarily correct) opinion, the latter point should have been tackled with some pros and cons already in chapter 5 and not just as the last sentences of the manuscript.

*Response:* We understand the reviewer's point here, but we thought it best (particularly for readers who might be less familiar with Arctic paleoceanography) to leave this text separated from the discussion, so that it will not get lost within detailed arguments about aspects of Arctic Ocean history.

Specific comments by line numbers
93: Actually, the isotopic change during the transition from sea water to sea ice is only very minor (fractionation factor ~1.003; https://doi.org/10.3189/S0022143000042751) and can be neglected when isotopic changes on glacial-interglacial scales are discussed. However, the d18O/salinity relation in ocean waters can be strongly affected. This can lead to density changes and the sinking of low-d18O near-surface waters to greater depths. Considering these details, the statement in line 93 is too much simplified.

*Response:* This is a good point; we updated the text on L95-102 to distinguish between the fractionation during brine formation (minimal) and the densification of low-d18O surface waters by brine addition.

109: What is "AL"?
*Response:* Changed to Atlantic Water (AW) throughout for consistency.

124: color bar
*Response:* Change made.

144: analysis. Analyses
*Response:* Change made.

185-186: Since the default R is 550 yr in Marine20 (and was 402 yr in Marine13), it might be worth mentioning the R used in this study.
*Response:* We specified ΔR=0 here.

186: Blaauw (to be corrected also in the list of references!)
*Response:* Spelling corrected.

192: Bulimina aculeata
*Response:* Spelling corrected.

200: Lomonosov Ridge
*Response:* Spelling corrected.

282: A temperature of -0.3°C may be correct for intermediate waters (AIW), but deeper waters are colder (-0.9°C; see line 106).
*Response:* Updated to "-0.3 to -0.9°C" to encompass the range of BWTs expected for our core sites.

284: Temperatures are numbers and cannot be "warmer", only higher. Check for other usage of "cooler" and "warmer" BWTs throughout the manuscript!
*Response:* Good catch – we changed all direct references to °C to "higher/lower" while keeping general comparisons of bottom waters as "warmer/cooler".

289-291: I suggest to discuss the differences going from 50 ka to present and not vice versa.
*Response:* We changed the presentation of $\delta^{18}O_b$, bottom water temperatures, and $\delta^{18}O_{sw}$ here to from oldest to youngest.

346 (and 437): I would regard 7b and 13b as interstadials within MIS 7 and 13, comparable to MIS 5d and 5b within MIS 5. Please note that 5b (which was globally just a colder interval within MIS 5) saw one of the largest glaciations over northern Eurasia in the last 200 ky (Svendsen et al., 2004, QSR).
*Response:* We specified stadials vs. glacials throughout the text.

375: Since freshwater is buoyant, wouldn't it make more sense to say that the Arctic Ocean may have been filled with freshwater DOWN to 2500 m water depth? It certainly depends on the point of view, but I would understand "up to 2500 m" as the description of a bottom water mass.
*Response:* Good point and agreed; we changed to "down to 2500 m water depth".

451: Spielhagen
*Response:* Spelling corrected.

455: timescales???
*Response:* Typo deleted.

470: Moreover, there were no large ice sheets on northern Eurasia during MIS 5c (Svendsen et al., 2004, QSR) to produce large amounts of brines.
*Response:* This is a good point, but we seek to be cautious here considering the simplicity of the age model. The one Nd isotope datapoint within MIS 5 on this age model dates to 94 ka (within MIS 5c), but the uncertainty on this age estimate is probably significant. We do not think we could rule out that this point overlaps with the expanded Kara-Barents Ice Sheet of the Early Weichselian.

519-521: This is correct. However, there is a recent paper (Rogge et al., 2022, https://doi.org/10.1038/s41561-022-01069-z) showing the formation of plumes which are sinking down to 1200 m even under modern conditions. During glaciations with (partly) ice-covered shelves and strong erosion, conditions allowing the formation of sediment-laden plumes may have occurred even more frequently than during interglacials. Since the Arctic deep-sea basins (>2500 m) are filled with fine-grained sediments (plumites, turbidites; see Goldstein, 1983, DOI: 10.1007/978-1-4613-3793-5_9; Svindland and Vorren, 2002, https://doi.org/10.1016/S0025-3227(02)00197-4), such plume formation may have been a major process for the vertical (and then horizontal) transport of both fines and low-d18O waters during glacials. Maybe the authors want to consider this possibility...

*Response:* This is an interesting observation and we have included the reference to Rogge et al. (2022) in the discussion of potential densification pathways for surface waters. However, we do not wish to place too much emphasis on this mechanism, as it appears (at least to us) difficult to reconcile an increase in sediment-laden plume transport to the deep Arctic during glacial intervals, when glacial sedimentation rates were lower than interglacial sedimentation rates. If anything, we might predict the opposite pattern based on sedimentation rates (more plume delivery during interglacials, because sedimentation rates were higher).

Figures: When figures consist of several "subfigures" (e.g., data panels), they are labeled A, B, C... In the text, they are referenced as a, b, c...

*Response:* We updated panel labels to lower case on figures and in the captions to fit Climate of the Past formatting.

**Comment on egusphere-2022-1212**
**Kaustubh Thirumalai (Referee)**
Referee comment on "A 600-kyr reconstruction of deep Arctic seawater δ18O from benthic foraminiferal δ18O and ostracode Mg/Ca paleothermometry" by Jesse R. Farmer et al., EGUsphere, https://doi.org/10.5194/egusphere 2022-1212-RC2, 2022

Kaustubh Thirumalai, University of Arizona

I found this manuscript by Farmer and colleagues to be highly interesting. The authors attempt to reconstruct bottom water d18Osw values in the Arctic using a combination of benthic foraminiferal d18O measurements paired with ostracode Mg/Ca paleothermometry. Using Site 1123 (SW Pacific Ocean) bottom water d18Osw as a reference record, they find that local Arctic d18Osw (corrected for the influence of ice volume-related changes in global oceanic d18O) is lower than the modern difference between these records during glacial periods over the past 600 kyr. A major portion of the discussion in this manuscript focuses on refuting the "fresh glacial Arctic" hypothesis (Geibert et al. 2021) to explain the anomalously lower d18Osw values in the glacial - which is compelling. The authors instead prefer a mechanism that involves "stratification breakdown" wherein relatively lower-d18O upper ocean waters sink to the bottom due to relatively higher densities (via salinity) modulated by brine formation.

Overall this work is compelling, of interest to the broader community, and I feel that the manuscript will be eventually suitable for publication in Climate of the Past pending some revision. My major concerns regarding this version are two-fold: one centers around the discussion and proposed mechanism of brine-formation to explain relatively lower-d18O in Arctic bottom-waters, and the other is a request to assist readers by providing more information on some of the details related to Mg/Ca paleothermometry and background information. I detail my major and minor suggestions to improve this work below:

Mechanisms and "low-d18O" of brine versus "lower d18O" of surface waters:
• The authors refer to "low-d18O" brines (Line 93), but the briny waters themselves are not anomalously lower in their d18Osw values due to relatively low ice-water d18O fractionation (see e.g. Yamamoto et al. 2001, 2002). Thus, I suggest expanding the text in the introduction to discuss how brine potentially affects bottom water d18O by advecting relatively lower upper-ocean d18Osw (Line 93 and thereafter).
This is a good point; we updated the text on L95-102 to distinguish between the fractionation during brine formation (minimal) and the densification of low-d18O surface waters by brine addition.

• I ask the authors to explore/discuss the work of Rasmussen and Thomsen (2009) who suggest that the initial thermohaline origin of brine formation can modulate their density and stable oxygen isotope composition. I recognize that the viewpoint from these authors on brine formation as a paleoceanographic driver has been updated since that paper (e.g. Rasmussen and Thomsen, 2014). However, their benthic d18O values in a region of active brine formation in the Barents Sea appear similar (Figs. 2–3 in 2009 paper) to the relatively lower glacial values observed in this manuscript (e.g. Fig. 3A). Perhaps this can be used as support for brine-formation as a potential cause of the underlying data? The authors also discount the possibility of reduced Atlantic water input during glacial stages.
*Response:* We expanded the discussion of brine formation in Section 5.2, but we note that our discussion of brines is necessarily simple because of the lack of available data constraints. We emphasize that, with the data constraints available, we cannot exclude that brine formation may have contributed to the intermediate-to-deep Arctic d18Osw patterns we observe, nor can we prove that brine formation had any impact on these d18Osw patterns. Our hope is that these d18Osw data provide motivation for the collection of additional datasets now outlined in the new Section 5.3 that can better refine the role of brine formation in the modification of Arctic deep waters.

• Although the records do not overlap entirely with the one presented in this manuscript, Ford and Raymo (2019) show that d18Osw (not corrected for ice volume) from ~400–600 ka at Sites 607 and 1208 in the North Atlantic and Pacific respectively are relatively similar to that at Site 1123 (Fig. 3 in their paper). Thus, perhaps the authors

can use these records across the interval of overlap to more robustly assert that changes in inflow did not cause the anomalously low glacial d18Osw in the Arctic Ocean (e.g. via a mixing model or relative differences between sites)? I recognize that their data points are fewer in this interval and that overlap is not complete, yet, I feel that this exercise would solidify their argument.

*Response:* Thank you for this idea. We added the Ford and Raymo DSDP 607 d18Osw reconstruction to Figure 4 and assess its comparability in the discussion.

• What is the driver of the stratification breakdown in the Arctic Ocean under glacial conditions? Presumably there was more perennial sea-ice coverage during glacial times, which could perhaps result in more year-round brine formation, but what would cause more mixing given that this would also likely reduce the impact of winds? Given that perennial Arctic sea-ice under pre-industrial conditions also covered a large extent of the basin, what changes during glacial times that cause radically different oceanic structure?

*Response:* We have updated Section 5.2 to better discuss the glacial stratification breakdown mechanism proposed by Farmer et al. (2021) (L499-516) and include a new figure (Figure 9) presenting the proxy data supporting this mechanism.

• On this note, I wonder whether the "hyperpycnal flows" hypothesis of Stanford et al. (2011) might have a role to play here? Could intensified hyperpycnal flows related to runoff/melt in spring/summer coupled with more brine formation in the fall/winters be a viable way to reduce year-round d18Osw in the water column? Stanford et al. (2011) also reject brine formation as a likely mechanism in the North Atlantic as there are different predictions for planktic versus benthic d18O values. Considering this, I suggest the authors to expand their discussion section to include the implications of (not-yet-generated) planktic foraminiferal derived/surface-ocean d18Osw in the Arctic and how it may help falsify/strengthen their hypothesis.

*Response:* We provided a new paragraph in Section 5.3 (L561-569) to discussing how planktic foraminiferal derived d18Osw could help clarify the cause(s) of lower Arctic d18Osw during glacials.

Discussion surrounding Mg/Ca and its uncertainty
• The authors ought to provide more background information on ostracode Mg/Ca analysis and their underlying uncertainties. Although it is mentioned that a 'Fisons Instrument Spectraspan atomic emissions spectrometer' instrument was used (Lines 157–158), no details are provided about instrument precision, matrix/standard effects, external standard replicability, numbers of valves analyzed, what instaars were utilized, inter-sample variance etc. Although the authors cite previous studies that go more into detail on some of these aspects, I think that these details need to be in this manuscript, where the Mg/Ca data are front and central to the assertions.

*Response:* We updated Section 3.2.2 (L154-166) to provide additional methodological details on the Mg/Ca determination.

• Accordingly, it is not clear how the "1sd" propagated error in Fig. 3A was constructed - was this propagated through a Monte Carlo procedure? It doesn't seem like a constant "calibration" error of ±1°C (Line 164–165; Farmer et al. 2012) was employed. Thus, a clearer discussion of the procedures employed to propagate Mg/Ca uncertainties into the resultant BWT and d18Osw records is needed.

*Response:* We agree that this was unclear, and for clarity in calculation, error propagation, and comparison to other studies (Sites 607 and 1123), we recalculated d18Osw for the Arctic record using PSU Solver (Thirumalai et al., 2016). The updated data and their uncertainties are plotted in Figures 3, 4, 7, 8, and 9 and included in the Supplementary Source Data.

• Relatedly, there is no discussion about other trace metal values such as Mn/Ca, Fe/Ca, or Al/Ca in these measurements, which were presumably also measured alongside Mg/Ca for investigating clay contamination. To what extent are these parameters correlated or uncorrelated with the Mg/Ca data and what are the implications for sediment-based or post-depositional alteration?

*Response:* On L161-164, we specified that Al, Fe, and Mn were monitored as indicators of clay contamination during Mg/Ca analysis.

Statistical robustness of d18Osw reconstruction:

• Although I am somewhat convinced looking at Fig. 4D that glacials generally have lower d18Osw in the Arctic relative to Site 1123, I feel that this can be done in a more robust manner by compositing glacials and interglacials across the entire record - what is the average value (my apologies if I missed this in the text) of the difference in glacials versus interglacials? What is the impact of highly variable Δd18Osw values in MIS 7, 11 and 13 interglacials on this average, as many of these anomalies appear to be equivalent or lower than the MIS 2–4 glacial values? Moreover, how robust is this average glacial-interglacial difference relative to the modern difference given the propagated uncertainty (see above point on Mg/Ca)? Although the 0.3–0.6 ‰ anomalies are likely significant, I wonder about smaller magnitude anomalies…

*Response:* This is a good point and is now addressed in Section 4.2, where interglacial and glacial Δd18Osw are tested for significance (L352-360).

Minor Points:
• It would be helpful to label in the caption that Fig. 4C refers to d18OL+IVC at both of the sites, right? If so, given that the record in Fig. 4D is constructed to be non-dependent on ice-volume/sea-level, I ask whether Fig. 4C is required at all? Moreover, this sub-plot is not cited/discussed explicitly in the text.

*Response:* We made two changes for clarification here: First, we explicitly defined and refer to ice-volume corrected d18Osw (as in Figure 3d) as d18Osw-ivc. Second, we expanded our discussion of Figure 4c in the text to compare Arctic d18Osw with d18Osw from Sites 607 and 1123.

• The authors provide information about the d18O measurements - however, am I to understand correctly that this work only involved collating previously measured d18O data? In that case, I would recommend adding "In these studies" before "Foraminifera were brush-picked," on Line 140 and updating Line 148 to say "In these previous studies, all measurements were reported…" to clarify what has been done in this work versus previous studies.

*Response:* Good point; we have adjusted accordingly.

• Line 225: Although the average error and standard deviation was relatively low, I'd recommend providing an estimate of the overall ranges as well (Eqn. 3 estimates a of range from X to Y whereas Eqn. 4)

*Response:* As we have now updated the methods to use PSU Solver, we removed this comparison from the text.